# OPEN-CK🔥: A LARGE MULTI-PHYSICS FIELDS COUPLING BENCHMARKS IN COMBUSTION KINETICS

**Zaige Fei**[*,1], **Fan Xu**[*,1], **Junyuan Mao**[1], **Yuxuan Liang**[2], **Qingsong Wen**[3],
**Kun Wang**[†,1,4], **Hao Wu**[†,1], **Yang Wang**[†,1]
[1] University of Science and Technology of China
[2] The Hong Kong University of Science and Technology (Guangzhou)
[3] Squirrel Ai Learning    [4] Nanyang Technological University
{zaige, markxu, maojunyuan, wuhao2022, angyan}@mail.ustc.edu.cn,
{yuxliang}@outlook.com,   {qingsongedu, sj675627966}@gmail.com

## ABSTRACT

In this paper, we use the Fire Dynamics Simulator (FDS) combined with the *supercomputer* support to create a **C**ombustion **K**inetics (CK) dataset for machine learning and scientific research. This dataset captures the development of fires in industrial parks with high-precision Computational Fluid Dynamics (CFD) simulations. It includes various physical fields such as temperature and pressure, and covers multiple environmental combinations for exploring multi-physics field coupling phenomena. Additionally, we evaluate several advanced machine learning architectures across our `Open-CK` benchmark using a substantial computational setup of 64 NVIDIA A100 GPUs: ❶ vision backbone; ❷ spatio-temporal predictive models; ❸ operator learning frameworks. These architectures uniquely excel at handling complex physical field data. We also introduce three benchmarks to demonstrate their potential in enhancing the exploration of downstream tasks: (a) capturing continuous changes in combustion kinetics; (b) a neural partial differential equation solver for learning temperature fields and turbulence; (c) reconstruction of sparse physical observations. The Open-CK dataset and benchmarks aim to advance research in combustion kinetics driven by machine learning, providing a reliable baseline for developing and comparing cutting-edge technologies and models. We hope to further promote the application of deep learning in earth sciences.

## 1 INTRODUCTION

Modern deep learning (DL) approaches have demonstrated promising outcomes in various dynamical systems in natural and social science fields like weather forecasting Schultz et al. (2021); Pathak et al. (2022); Bi et al. (2022), rapid fire progression Tam et al. (2022), and intelligent transportation Kaffash et al. (2021); Jin et al. (2023). Such astonishing achievements primarily stem from two crucial factors. First, with the development of computer science, a vast amount of data from Earth systems is continuously being acquired Chen et al. (2022); Liu et al. (2023). These ever-growing, massive datasets, with diverse sources, provide the impetus for data-hungry deep models, making learning from data possible. Second, continual breakthroughs in DL algorithms and models enable us to effectively adapt to diverse specific scenarios, resulting in state-of-the-art performances Wu et al. (2024a); Wang et al. (2024); Wu et al. (2023b; 2024c); Hao et al. (2024).

Fluids, an essential data type within the realm of earth sciences Ferziger et al. (2019); Temam (2001), are characterized by a molecular structure that lacks resistance to external shear forces. This inherent property allows fluids to deform readily, even under minimal forces, often resulting in dynamics that are highly complex both spatially and temporally Ma et al. (2024); Yu et al. (2018); Wang et al. (2021; 2022a). To date, an ever-increasing focus on data-driven deep methods, while paving

---

[*]Equal contribution
[†]Corresponding author

**Figure 1:** The figure shows the changes in different physical fields during an industrial park fire simulation, including the *pressure* field, *temperature* field, *velocity* field, and a colored *streamplot* of the velocity field. The simulation conditions are a single fire source and wind direction (10 mW combustion power, 1 m/s wind speed, 180 seconds simulation time)

a potential path for large-scale fluid dynamics studies, unfortunately poses a daunting obstacle in collecting high-quality fluid data Chen et al. (2022); Veillette et al. (2020).

A quick review of existing literature reveals that numerous benchmarks involving *sensor collection* and *numerical simulations* related to fluid dynamics have been proposed. In the field of meteorology, datasets such as RainNet Ayzel et al. (2020), ERA5 Muñoz-Sabater et al. (2021), and WeatherBench Rasp et al. (2020) have collected high-quality meteorological data. These resources offer detailed measurements of rainfall, hurricanes, temperature, climatic variables related to land and oceans, which provide invaluable resources for subsequent meteorological model development like Pangu-Weather Bi et al. (2022). In the ocean domain, HYCOM Chassignet et al. (2007), ECCO Forget et al. (2015), and CMEMS provide detailed measurements of ocean temperature, salinity, currents, and sea surface height, offering valuable data for ocean model development. Going beyond this process, Many fluid benchmarks across terrestrial environments Justice et al. (2002;?); Rodell et al. (2004) and spring systems Otness et al. (2021) provide foundational research platforms for this field Wu et al. (2023b).

Though promising, combustion safety, which is crucial for human life, lacks high-quality standardized datasets and evaluation protocols. Existing combustion dynamics datasets like WildfireDB Gerard et al. (2024), FireSpread Gerard et al. (2024), and Prometheus Wu et al. (2024b) have limitations. WildfireDB and FireSpread mainly rely on actual image data, which limits diversity and coverage and lacks physical significance. These datasets also face data imbalance and noise issues, affecting model training. While Prometheus provides high-quality data, its low spatial and temporal resolution fails to capture complex fire dynamics. Additionally, these datasets lack multi-physics coupling data, limiting model generalization across scenarios. Therefore, we develop the Open-CK Benchmark. Open-CK (🔥) is the underline{first} open-source benchmark dedicated to the study of combustion fluid dynamics, created through over 360 hours of numerical simulations supported by `supercomputers`. This dataset captures the ST variations of **multiple** physical fields (as shown in Figure 1), encompassing 300 scenarios with varying parameters such as heat release rates (HRR) of Standards & Technology (2023), temperature, source location, *etc*, contributing to the development of a standard evaluation protocol in fire safety research.

Open-CK involves several PDEs, including the Navier-Stokes Li et al.; Takamoto et al. (2022), mass conservation Jain & Kennedy (2014), energy conservation McGrattan et al. (2010), the Heat Conduction Tieszen (2001), and the Transport Equation for Smoke and Chemical Species Drysdale (2011). The Fire Dynamics Simulator (FDS) Hietaniemi et al. (2004), a computational fluid dynamics (CFD) Hirsch (2007) software, leverages these PDEs to simulate the complex dynamics of combustion McGrattan et al. (2006), including flame propagation, smoke flow, heat conduction, and radiation. By solving these equations, FDS provides detailed simulations and analyses of fire scenarios. Using FDS, we generated ∼480GB of raw data in CSV format, organized in a two-dimensional (time-sensor) structure. To facilitate research using 🔥, we preprocessed this data into a four-dimensional ($\mathcal{D}_1 \sim \mathcal{D}_4$) format: $\mathcal{D}_1$ represents sampling time, $\mathcal{D}_2$ the type of physical field (velocity, temperature, pressure), and $\mathcal{D}_3$ and $\mathcal{D}_4$ represent the two-dimensional spatial distribution of sensors. This preprocessing compresses the raw data into a more manageable size (∼2.2T) and formats it into npy files Harris et al. (2020) suitable for DL applications.

**Boarder Impact.** 🔥 has the following features and advantages: Firstly, we use FDS simulations to generate various industrial park fire scenarios. Unlike machine learning benchmark

PDEBENCH's Takamoto et al. (2022) single physical field, Open-CK covers **multiple physical** fields, including temperature, pressure, and velocity, with high spatial and temporal resolution. Secondly, compared to large CFD-based fire set EAGLE Janny et al. (2023) and Prometheus's Wu et al. (2024b) single grid environments, 🔥, supported by `supercomputers`, simulates grids of different sizes, ensuring data complexity and high physical consistency. Lastly, we validate the dataset's effectiveness through multiple SOTA models. By comparing various advanced machine learning models, we demonstrated the dataset's potential in combustion kinetics research. Our study not only provides a scientific basis for fire prediction and management but also opens new avenues for the application of deep learning in Earth sciences.

## 2 PRELIMINARIES

### 2.1 PROBLEM FORMULATION

This study explores the ST evolution of fire dynamics, where the input data is represented as a four-dimensional array $[T, C, H, W]$. $T$ denotes the time steps, $C$ represents channels for different physical values (*e.g.*, temperature $\mathcal{R}$ and pressure $\mathcal{V}$), and $H$ and $W$ correspond to the vertical and horizontal spatial resolutions, respectively. To predict future combustion dynamics behavior, we can utilize a model based on historical data, employing Maximum Likelihood Estimation (MLE) for parameter estimation. Let the model output $Y_t$ represent the state of combustion at time $t$, and $X_t$ contains all prior time step data, i.e., $X_t = [x_1, x_2, \ldots, x_t]$. Our goal is to estimate $Y_{t+1}$, the state at the next time step, through the predictive model $f(\cdot)$. The mathematical representation is as follows:

$$Y_{t+1} = f(X_t, \Theta), \quad \mathcal{L}(\Theta; X_t, Y_{t+1}) = \prod_{i=1}^{T} P(y_{t+1}^i | x_t^i; \Theta) \quad (1)$$

where $\Theta$ is the model parameter. Using MLE, we aim to find the parameters $\Theta$ that maximize the likelihood $P(Y_{t+1}|X_t; \Theta)$ given $X_t$. By maximizing the log-likelihood function $\mathcal{L}$, an estimate of the parameters $\Theta$ can be obtained as $\hat{\theta} = \arg\max_\theta \log \mathcal{L}(\theta; X_t, Y_{t+1}) = \arg\max_\theta \sum_{i=1}^{T} \log P(y_{t+1}^i | x_t^i; \Theta)$. Scrutinizing above optimization, the model can not only effectively utilize $X_t$ to predict future combustion dynamics $Y_{t+1}$ but also perform predictions over multiple time steps, providing a scientific basis for fire management and control.

### 2.2 TECHNICAL BACKGROUND

**Scientific Machine Learning Datasets.** Multiple efforts aim to develop benchmark datasets for scientific machine learning. WeatherBench Rasp et al. (2020; 2023) offers global high-resolution weather forecasting data, covering meteorological variables like temperature, pressure, and wind speed. The ERA5 dataset Muñoz-Sabater et al. (2021), organized by the European Centre for Medium-Range Weather Forecasts (ECMWF), has provided hourly estimates of atmospheric, land, and oceanic climate variables since 1940, becoming a popular resource for weather prediction. It has supported the development of neural weather models like FourCastNet Pathak et al. (2022), Pangu-Weather Bi et al. (2023), and Graphcast Lam et al. (2022). PDEBENCH Takamoto et al. (2022) suite for scientific machine learning models dealing with systems governed by PDEs, which provides diverse initial and boundary conditions and introduces new metrics to assess scientific performance. Primarily focused on time-dependent flow problems, PDEBENCH does not include complex scenarios like multi-phase flows or non-rectangular domains. Large benchmark EAGLE Janny et al. (2023) contains $\sim 1.1$ million 2D grids simulated based on Computational Fluid Dynamics (CFD) techniques, generated by simulations of non-constant hydrodynamics induced by the interaction of a moving flow source with a nonlinear scene structure; however, all snapshots are too low-resolution for the analysis of complex phenomena. WildfireSpreadTS Gerard et al. (2024) is a multi-temporal and multi-modal dataset with 13,607 images from 607 U.S. wildfires from 2018 to 2021. It supports high-resolution predictions of wildfire spread using its time series structure and multi-modal inputs. However, the dataset faces challenges due to complex inputs, imbalanced labels, and noisy data.

**Dynamic System Modeling (DSM).** Various advanced architectures that each offer unique features for addressing complex problems in DSM realm. ❶ Vision backbone architectures like U-Net and its variants Ronneberger et al. (2015); Huang et al. (2020); Li et al. (2018); Weng et al. (2019), with their

symmetrical encoder-decoder structure, optimize feature extraction from high-resolution images, making them ideal for super-resolution tasks in meteorological data. ResNet He et al. (2016) introduces residual connections to combat gradient vanishing, enhancing learning capabilities. Vision Transformer (ViT) Dosovitskiy et al. (2021) effectively captures global image dependencies through its self-attention mechanism, suited for large-scale, complex physical field data Pathak et al. (2022). ❷ In spatio-temporal prediction realm, ConvLSTM Shi et al. (2015) merges the strengths of convolutional networks and long short-term memory networks to handle spatial correlations and time series data efficiently. Earthformer Gao et al. (2022b) leverages the Transformer's encoding power for large-scale earth science data processing. SimVP Tan et al. (2022) efficiently predicts future states of dynamic systems with a simplified architecture. ❸ In operator learning field, architectures like Fourier Neural Operator (FNO) Li et al. (2020), U-shaped Neural Operators (UNO) Ashiqur Rahman et al. (2022), and Convolutional Neural Operator (CNO) Raonic et al. (2024) innovate in Fourier space, multi-scale representations, and convolutional structures, respectively, offering precise and universal solutions for solving partial differential equations. These architectures advance dynamic system modeling and expand opportunities for scientific research and industrial applications.

# 3 Open-CK: The Non-linear Chaotic Combustion Kinetics Benchmark

In this section, we formally introduce the `Open-CK` benchmark, designed to comprehensively assess the accuracy, efficiency, and fidelity of existing and future vision backbones, ST predictive models, and neural operator frameworks. Initially, we provide a detailed explanation of the collection and organization of Open-CK in Section 3.1. In Section 3.2, we introduce the dataset management and workflow, followed by an in-depth explanation of further preprocessing steps to enhance our understanding of Open-CK. Subsequently, in Section 3.3, we analyze the data distribution of Open-CK and present the characteristics of the dataset. Finally, in Section 3.4, we outline the licensing of the resources.

## 3.1 Data Collection and Organization

To effectively train deep learning models for modeling and predicting fire scenarios in industrial parks, we collect a large set of simulated data. We generate this data using version 6.9.1 of the FDS developed by the National Institute of Standards and Technology (NIST), based on CFD simulations. The simulations occur in a typical industrial park with multiple oil storage areas, modeling various fire development scenarios. The simulated scenario appears as part (I) in Figure 2(a). We create 300 different fire scenarios, each with specific settings for fuel type, heat release rate (HRR), fire growth factor, wind speed, wind direction, and ignition location. We run all simulations on a `supercomputer`, with each scenario taking approximately seven to ten hours. To accurately capture the dynamic changes in the physical field during fires, we place a dense sensor grid at a height of five meters within the simulation area, with sensors spaced every 0.5 meters. The sensor arrangement appears as part (II) in Figure 2(a). These sensors collect data every second, covering multiple physical parameters such as velocity, temperature, and pressure, as detailed in Table 1. We organize the collected data into time series, with each data point containing the full physical parameters of the scene at each moment, ensuring data integrity and continuity. We then format these data into a structure suitable for deep learning model input, providing a high-quality, dense numerical database for algorithm training and validation. This approach not only allows us to study the physical behaviors in various fire scenarios in detail but also to optimize and adjust the deep learning models based on these simulation results, improving their application in real-world fire prediction and management.

AutoCAD[1] is a widely used computer-aided design (CAD) software for precise creation of both 2D and 3D drawings and models. We used AutoCAD software to create a 3D model of an industrial park in a real-world scenario (as shown in Figure 9). The resulting DWG files[2] were then imported into PyroSim software[3] for the preliminary design work required for numerical simulation (as shown in Figure 12). PyroSim is a software tool commonly used for fire and evacuation simulation in

---

[1] https://www.autodesk.com/products/autocad/overview

[2] DWG (short for "drawing") is a proprietary binary file format used for storing two- and three-dimensional design data and metadata. https://www.autodesk.com/solutions/dwg

[3] https://www.thunderheadeng.com/pyrosim/

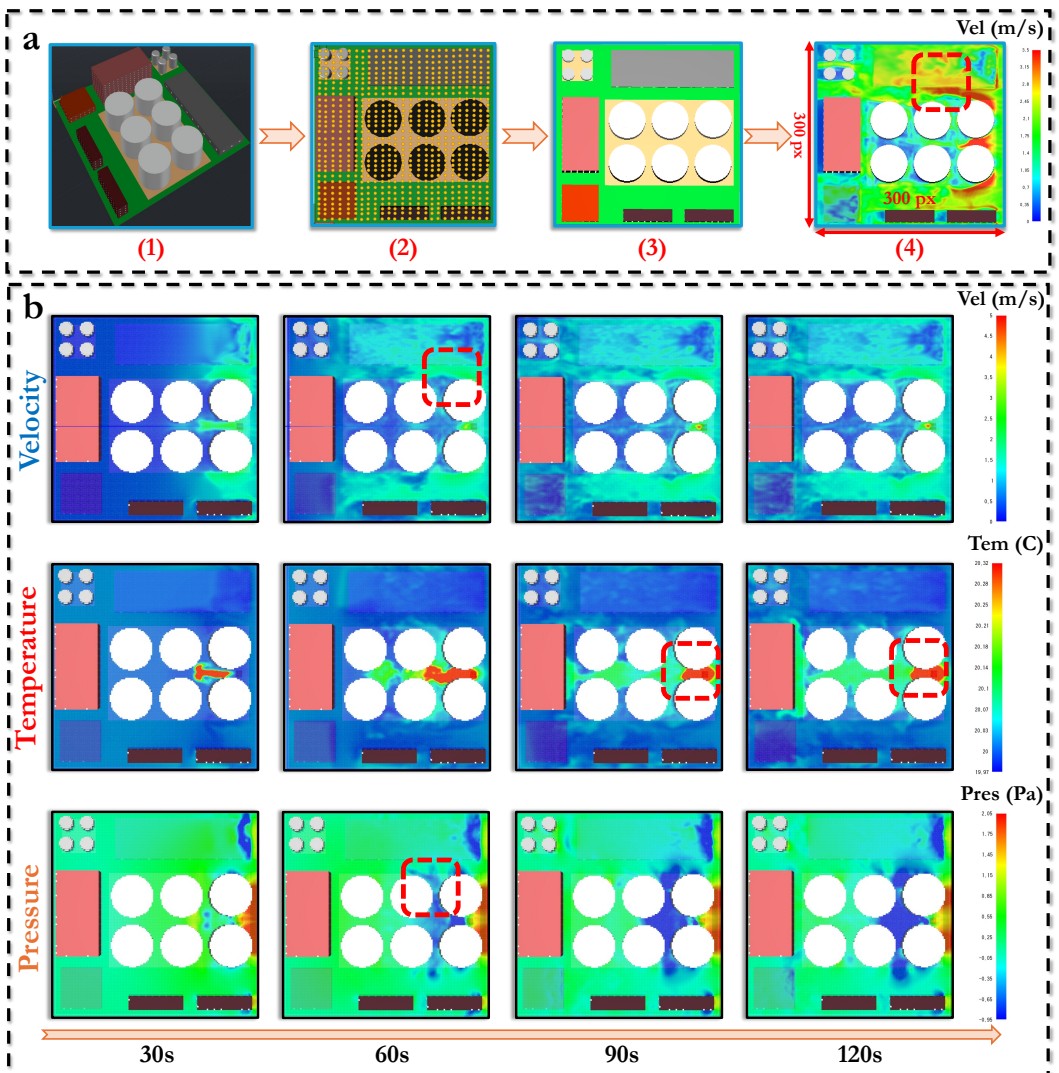

**Figure 2: a** describes the overall layout of a simulated industrial park experiment. **b** shows visualizations of different physical fields changing over time.

complex environments. This design work includes setting parameters such as mesh size and density, fire source location and type, ventilation conditions, HRR, and fire growth coefficient. This process generates the FDS files[4] corresponding to the various scenarios.

Python[5] is a widely used programming language. Due to the large number and variety of sensors needed, we wrote a Python script (as shown in Figure 14) to directly edit the FDS files and create the required sensor arrays. After running the FDS files for numerical simulation, we obtained the data collected by all the sensors. This data is saved in CSV format, *where the columns represent all sensors and the rows represent all recorded time points.* For illustration, in a particular scenario, we set up three types of sensors, each with 90,000 units. This scenario simulated the fire development process over 400 seconds, with a data collection frequency of 1 Hz. Thus, the resulting CSV file has dimensions of (400, 270000). Additionally, each numerical simulation generates an SMV file[6],

---

[4]FDS files are input files for the Fire Dynamics Simulator, a CFD model of fire-driven fluid flow. These files contain data on the geometry, material properties, and initial conditions needed to simulate fire scenarios. More information can be found at https://pages.nist.gov/fds-smv/

[5]https://www.python.org/

[6]An SMV file is a visualization file format used by Smokeview, a companion software to the FDS.

**Table 1:** Summary of Open-CK. NIS: Number of Ignition Sources; SIS: Single Ignition Source; TIS: Three Ignition Sources; Tem: Temperature; Vel: Velocity; Pres: Pressure.

| NIS | Physical Field | Domain | Spatial Resolution | $\Delta t$ | Timesteps |
|-----|----------------|--------|--------------------|-----------|-----------|
| SIS | Tem&Vel&Pres | 300m x 300m | 300 x 300 | 1 | 350 |
| SIS | Vel | 110m x 70m | 220 x 140 | 1 | 300 |
| SIS | Tem | 20m x 20m | 40 x 40 | 0.5 | 250 |
| SIS | Pres | 20m x 20m | 20 x 20 | 2 | 350 |
| TIS | Tem&Vel&Pres | 300m x 300m | 600 x 600 | 1 | 350 |
| TIS | Tem | 110m x 70m | 220 x 140 | 1 | 300 |
| TIS | Vel | 50m x 50m | 50 x 50 | 0.5 | 300 |
| TIS | Pres | 20m x 20m | 40 x 40 | 2 | 350 |

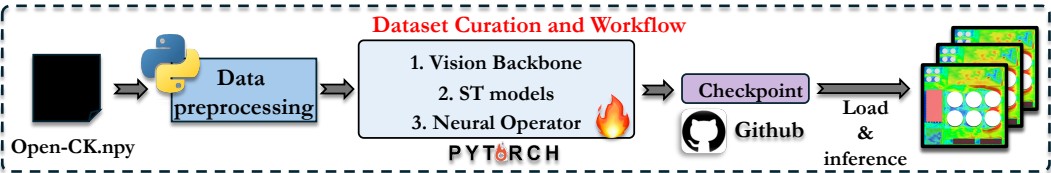

**Figure 3: Dataset Curation and Workflow.** This figure shows the process from data preprocessing to model training, saving, and application. Data is preprocessed using Python, analyzed and features extracted via a vision backbone, ST models, and a neural operator in PyTorch. The trained model is saved as a checkpoint, uploaded to GitHub for storage and version control, and used for inference.

which can be visualized using PyroSim's PyroSim Results application, a feature within the PyroSim software designed for viewing, analyzing, and interpreting simulation results (Figure 13).

To preprocess the data for use in deep learning models, we used Python scripts (see Figure 15) to filter and clean the data in the CSV files. The data was then reshaped to dimensions (T, Dt, R, C), where T represents the time steps, Dt represents sensor types, R represents the sensor rows, and C represents the sensor columns. Thus, we obtained the preliminary preprocessed data in NPY format. Further data processing, such as applying a sliding window, is detailed in Section 3.2.

### 3.2 DATASET CURATION AND WORKFLOW

Figure 3 shows a comprehensive data processing and workflow for a combustion dynamics physical field dataset. Initially, we use Python tool to handle the data preprocessing, including cleaning and formatting to enhance data quality and adaptability. Subsequently, the process employs foundational visual models, spatio-temporal models, and neural operators for in-depth data processing and feature extraction. These models train within the PyTorch framework and save as checkpoints for storage and further use. Finally, the workflow uploads these model checkpoints to GitHub for sharing and version control, enabling model loading for further inference and applications. During the data pre-processing phase, as shown in Appendix (see Figure 8), the process uses the Numpy library in PyTorch for numerical computations. It loads data from multiple files and concatenates it along a specified axis. The process defines a `sliding_window_view` function using the `numpy.lib.stride_tricks.as_strided` method to generate a sliding window view of the data. This method alters the array's strides to view different parts of the array without copying the original data, ideal for time series analysis. It captures time-dependent characteristics effectively and facilitates further analysis and model training, particularly when considering the data's temporal continuity and local features.

### 3.3 DATA ANALYSIS & CHARACTERISTICS

Figure 4(a) shows the distribution of temperature, velocity, and pressure and their changes. The temperature distribution shows most values are low with a big right skew. Over time, the temperature goes up, peaks, and then stays stable. The velocity distribution shows most values are low, and the time series shows a slow rise, leveling off later. The pressure distribution shows most values are in the middle with some extremes, and the time series shows a quick drop at first, then stability. Overall, these variables show clear phases over time, reflecting the system's initial changes and final stable state. **This highlights the advantages of our Open-CK dataset, which encompasses multiple physical variables.** Figure 4(b) shows the t-SNE clustering results for temperature, velocity, and

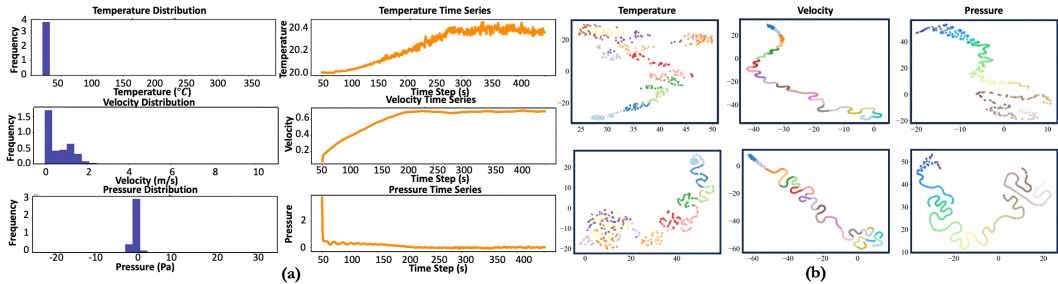

**Figure 4:** (a) Distribution and time series of temperature, velocity, and pressure. (b) t-SNE clustering results for temperature, velocity, and pressure.

**Table 2:** Baseline results across the Open-CK dataset. We make the best performance baseline by using ❶. ▨, ▨ and ▨ denote spatio-temporal, vision and neural operator backbones, respectively.

| Method | Params (M) | FLOPs (G) | FPS | MSE ↓ | MAE ↓ | SSIM ↑ | PSNR ↑ |
|---|---|---|---|---|---|---|---|
| ConvLSTM Shi et al. (2015) | 17.2301 | 60.4823 | 109.8573 | 0.0307 | 92.1234 | 0.9271 | 21.8497 |
| PredRNN Wang et al. (2017) | 25.3652 | 119.9854 | 53.2781 | 0.0243 | 73.9876 | 0.9445 | 23.0932 |
| PhyDNet Guen & Thome (2020) | 35.9784 | 169.9863 | 40.1256 | 0.0224 ❶ | 70.0154 ❶ | 0.9502 | 23.7154 ❶ |
| MIM Wang et al. (2019) | 39.0347 | 179.9743 | 36.2458 | 0.0232 | 70.4567 | 0.9478 | 23.4789 |
| PredRNNv2 Wang et al. (2022b) | 50.0789 | 299.9823 | 16.9832 | 0.0364 | 80.0234 | 0.9296 | 21.0032 |
| PastNet Wu et al. (2023c) | 4.0023 | 15.9874 | 179.8764 | 0.0282 | 79.0145 | 0.9363 | 22.4876 |
| SimVP Gao et al. (2022a) | 4.9873 | 17.9856 | 200.2345 | 0.0271 | 78.4967 | 0.9375 | 22.6045 |
| SimVPv2 Tan et al. (2022) | 23.9765 | 117.9823 | 50.7896 | 0.0246 | 73.9872 | 0.9448 | 23.2012 |
| Earthfarseer Wu et al. (2024a) | 24.0123 | 118.1234 | 51.0987 | 0.0245 | 73.9234 | 0.9446 | 23.1987 |
| ResNet He et al. (2016) | 60.0987 | 19.9876 | 209.8765 | 0.0321 | 89.9987 | 0.9248 | 21.7967 |
| U-Net Ronneberger et al. (2015) | 45.0987 | 17.2345 | 279.9876 | 0.0252 ❶ | 72.0034 ❶ | 0.9463 ❶ | 23.1984 ❶ |
| ViT Dosovitskiy et al. (2020) | 47.0012 | 17.0987 | 280.1234 | 0.0273 | 77.9823 | 0.9401 | 22.7989 |
| Swin Transformer Liu et al. (2021) | 46.9876 | 16.9874 | 290.2345 | 0.0362 | 96.1456 | 0.9137 | 21.5987 |
| MLP-Mixer Tolstikhin et al. (2021) | 47.0543 | 16.9845 | 289.9876 | 0.0359 | 96.0765 | 0.9143 | 21.5984 |
| FNO Li et al. (2020) | 51.0987 | 19.2345 | 189.8765 | 0.0298 | 79.5432 | 0.9364 | 22.2834 |
| LSM Wu et al. (2023a) | 59.0012 | 19.9987 | 209.8765 | 0.0323 | 89.5432 | 0.9249 | 21.7965 |
| CNO Raonic et al. (2024) | 45.0001 | 17.0032 | 280.7654 | 0.0251 ❶ | 72.0123 ❶ | 0.9458 ❶ | 23.2087 ❶ |
| UNO Ashiqur Rahman et al. (2022) | 47.9765 | 18.0012 | 280.6543 | 0.0274 | 78.0876 | 0.9402 | 22.7896 |
| NMO Wu et al. (2024d) | 46.9876 | 17.1098 | 290.7654 | 0.0361 | 95.9345 | 0.9142 | 21.6012 |

pressure. Temperature data points cluster closely in several distinct groups. Velocity and pressure data points display more complex and dispersed patterns, indicating more diverse variations in these variables over space or time.

### 3.4 OPEN-CK LICENSE

The LargeST benchmark dataset is released under a CC BY-NC 4.0 International License: https://creativecommons.org/licenses/by-nc/4.0. Our code implementation is released under the MIT License: https://opensource.org/licenses/MIT. The license of any specific baseline methods used in our codebase should be verified on their official repositories.

## 4 EXPERIMENT AND ANALYSIS

### 4.1 EXPERIMENTAL SETTINGS

**Dataset.** We select partial data for our main experiment. Specifically, with a heat release rate of 5MW, a single fire source, and one wind direction, we simulate wind speeds of 1m/s, 2m/s, 3m/s, 4m/s, and 5m/s. Using the $t^2$ fire growth model with a growth coefficient of 0.178, we calculate the time to reach steady-state fire as 167.6 seconds. Thus, we choose a simulation duration of 300 seconds for these conditions. Through numerical simulation, we obtain temperature data during the fire evolution, which we use as the original training and testing dataset.

**Baselines & Implementation Details.** We select representative models from three domains as baselines. ▷ **Spatio-temporal Predictive Models.** We feature ConvLSTM Shi et al. (2015), PredRNN Wang et al. (2017), PhyDnet Guen & Thome (2020), MIM Wang et al. (2019), PredRNNv2 Wang et al. (2022b), PastNet Wu et al. (2023c), SimVP Gao et al. (2022a), SimVPv2 Tan

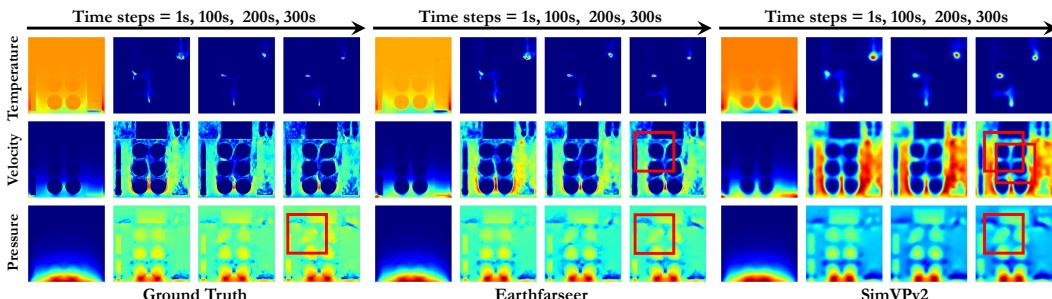

**Figure 5:** Comparison of predicted and true values of temperature, velocity, and pressure fields at different time steps (1s, 100s, 200s, 300s).

et al. (2022), Earthfarseer Wu et al. (2024a). ▷ **Vision Backbone.** We include ResNet He et al. (2016), U-Net Ronneberger et al. (2015), Vision Transformer(ViT) Dosovitskiy et al. (2021), Swin Transformer(SWINT) Liu et al. (2021) and MLP-Mixer Tolstikhin et al. (2021). ▷ **Operator Learning Frameworks.** We cover FNO Li et al. (2020), LSM Wu et al. (2023a), UNO Ashiqur Rahman et al. (2022), CNO Raonic et al. (2024), and NMO Wu et al. (2024d). Baseline details can be found in Appendix B. All backbones in this paper train with MSE loss, use the ADAM optimizer Kingma & Ba (2014), and set the learning rate to $10^{-3}$. The batch size is 50, and training early stops within 500 epochs. We train on 64 NVIDIA 40G-A100 GPUs. More details settings are left in Appendix C.

**Metrics Details.** We comprehensively evaluate the performance of supported models in the above tasks using different metrics based on task characteristics: **Error Metrics:** We use Mean Squared Error (MSE) and Mean Absolute Error (MAE) to measure the difference between predicted results and true targets. For weather forecasting, we typically use Root Mean Squared Error (RMSE). **Similarity Metrics:** We use Structural Similarity Index (SSIM) and Peak Signal-to-Noise Ratio (PSNR) to assess the similarity between predicted results and true targets. These metrics are widely used in image processing and computer vision. **Perceptual Metrics:** We use Learned Perceptual Image Patch Similarity (LPIPS) to evaluate the perceptual difference between predicted results and true targets in the visual system. LPIPS aligns with human visual perception and is suitable for fire video prediction tasks. **Physical Metrics:** We use Energy Spectrum Error to evaluate the physical consistency of deep learning models by calculating the energy spectrum error of the velocity field, assessing the model's performance in capturing fluid dynamics features. **Computational Metrics:** We assess the computational complexity of the model through the number of parameters and Floating Point Operations (FLOPs). Additionally, we report the frames per second (FPS) on a single NVIDIA A100 GPU to evaluate inference speed.

## 4.2 MAIN RESULTS

As shown in Table 2, the baseline results on the Open-CK dataset show various models performing differently across evaluation metrics. In spatio-temporal prediction models, PhyDNet excels with the lowest MSE of 0.0224 and MAE of 70.0154, indicating high accuracy in fire dynamics prediction. It also achieves the best SSIM of 0.9502 and PSNR of 23.7154, showing significant advantages in capturing complex spatio-temporal dynamics. PredRNN and SimVP have better FPS at 53.2781 and 200.2345, respectively, but their error metrics are not as strong. In vision backbones, U-Net performs best with MSE of 0.0252, MAE of 72.0034, SSIM of 0.9463, and PSNR of 23.1984, demonstrating excellent capability in high-resolution image processing. ResNet and ViT show high FPS at 209.8765 and 280.1234 but do not match U-Net in error metrics. In neural operator frameworks, CNO stands out with MSE of 0.0251, MAE of 72.0123, SSIM of 0.9458, and PSNR of 23.2087, proving its effectiveness in solving PDE problems. FNO and UNO also perform well in computational efficiency and some error metrics but are slightly inferior to CNO overall. These results indicate different models have unique strengths in handling combustion dynamics data. Spatio-temporal models like PhyDNet excel in accuracy and dynamic changes. Vision backbones like U-Net are outstanding in image processing. Neural operators like CNO show significant effectiveness in PDE solutions. This analysis provides a solid basis for optimizing and selecting backbones for specific tasks.

Additionally, the qualitative analysis results are shown in the Figure 5, comparing the predicted and true values of temperature, velocity, and pressure fields at different time steps (1s, 100s, 200s, 300s). The temperature field predictions are accurate at the initial time (1s) for all models, but Earthfarseer and SimVPv2 show significant deviations at 200s and 300s, especially at the boundaries of high-temperature regions. Velocity field predictions are good in the early stages (1s and 100s) but show large errors in the later stages (200s and 300s), with poor handling of high-velocity details. Pressure field predictions are accurate initially but have significant errors in high-pressure regions at 200s and 300s. Overall, Earthfarseer and SimVPv2 need improvement in long-term predictions and complex scenarios.

## 4.3 PHYSICAL CONSISTENCY ANALYSIS

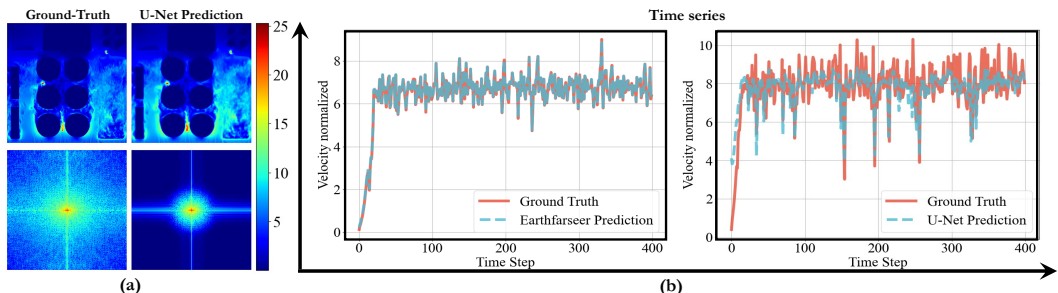

**Figure 6: Visualization of U-Net and Earthfarseer model performance in predicting velocity fields.** (a) The first row shows ground-truth and U-Net predictions, indicating high accuracy. The second row shows energy spectra, with U-Net closely matching the true spectrum but differing in high-frequency regions. (b) Time series plots of normalized velocity at arbitrary coordinates for all time steps show Earthfarseer and U-Net predictions align well with the ground truth. U-Net demonstrates slightly better consistency and accuracy in maintaining velocity trends.

Because the Open-CK benchmark has physical properties, we analyze its physical consistency Wu et al. (2024d); Wang et al. (2020) across different backbones in this section. The Figure 6 shows the performance of U-Net and Earthfarseer models in predicting velocity fields. In the first row of sub-figure (a), we see the ground-truth and U-Net prediction results. U-Net's output closely matches the ground-truth, indicating high accuracy. The second row shows the energy spectra. While U-Net's predicted energy spectrum is close to the true spectrum, there are some differences in the high-frequency region, indicating U-Net's slight deficiency in capturing fine structural changes. Subfigure (b) shows normalized velocity time series at arbitrary coordinates for all time steps. Earthfarseer and U-Net predictions align well with the ground truth. U-Net shows slightly better consistency and accuracy in maintaining velocity trends over time. This analysis highlights U-Net's effectiveness in high-fidelity spatio-temporal predictions for dynamic systems while also indicating room for improvement in high-frequency detail prediction.

## 4.4 CASE STUDY

In this section, we use temperature data from a three-source fire scenario for training and show relevant learning cases. Figure 7 shows the comparison of prediction results from Earthfarseer, U-Net, and FNO models with the ground truth. U-Net's predictions are very close to the ground truth, showing high accuracy. Earthfarseer and FNO capture the overall structure but have larger deviations in high-intensity areas. Table 3 lists the quantitative evaluation metrics for each model, including SSIM, LPIPS, and MSE. U-Net achieves the highest SSIM (0.9332), indicating the best structural similarity. It also records the lowest LPIPS (9.8632) and MSE (0.0259), reflecting better perceptual quality and lower prediction error. Earthfarseer also performs well, with an SSIM of 0.9217, LPIPS of 10.022, and MSE of 0.0267. FNO captures the general trend but performs slightly worse in SSIM (0.8977), LPIPS (11.2837), and MSE (0.0357). This analysis highlights U-Net's superior performance in both visual and quantitative evaluations.

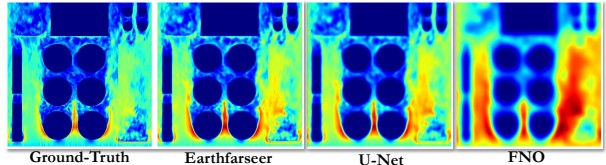

| Model | SSIM | LPIPS | MSE |
|---|---|---|---|
| Earthfarseer | 0.9217 | 10.022 | 0.0267 |
| U-Net | 0.9332 | 9.8632 | 0.0259 |
| FNO | 0.8977 | 11.2887 | 0.0357 |

**Figure 7:** Comparison of prediction results from Earthfarseer, U-Net, and FNO models with the ground truth.

**Table 3:** Quantitative evaluation metrics for Earthfarseer, U-Net, and FNO models, including SSIM, LPIPS, and MSE.

## 5 FUTURE INSIGHT & LIMITATION

To advance the field of fire dynamics research, we introduce the Open-CK as a new benchmark. This dataset integrates a total of 300 different fire scenarios simulated using high-resolution Computational Fluid Dynamics. It includes data from various physical parameters such as temperature, pressure, and velocity, providing a robust platform for comprehensive data analysis and modeling.

**Enhanced Model Generalization through Multi-Physics Coupling.** Our analysis suggests that combining data from multiple physical phenomena can significantly improve the prediction accuracy and robustness of fire dynamics models. Future research could leverage this multi-physics dataset to develop models that provide deeper insights into the complex interactions within fire dynamics, enhancing both predictive power and real-world applicability.

**A Testbed for Non-linear and Chaotic Combustion Dynamics.** Open-CK captures a wide range of non-linear and chaotic behaviors in fire scenarios, making it an excellent resource for testing new theories and models that address rapid changes in environmental conditions. This is especially valuable for developing strategies to predict and mitigate sudden fire escalations.

**Development of Real-Time Predictive Models.** Open-CK's extensive range and diversity provide an excellent foundation for developing real-time predictive models. This could potentially forecast the progression of fire and other dynamic phenomena as they happen, allowing for timely interventions.

While Open-CK is a powerful benchmark for fire dynamics research, it is not without its limitations. Firstly, Open-CK mainly uses simulated data that might not capture all the unpredictable changes seen in real-world fire events. Additionally, the accuracy of the simulations depends heavily on the fidelity of the input parameters and the resolution of the data, which can be compromised by computational limitations and sensor inaccuracies.

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

## A  INTRODUCTION TO PSEUDO ALGORITHMS

```python
import numpy as np
from numpy.lib.stride_tricks import as_strided

file_names = ['400x300x300-1.npy', '400x300x300-2.npy',
              '400x300x300-3.npy', '400x300x300-4.npy',
              '400x300x300-5.npy']

data = np.load(file_names[0])

for file_name in file_names[1:]:
    next_data = np.load(file_name)
    data = np.concatenate((data, next_data), axis=0)

def sliding_window_view(arr, window_size, step=1):
    n = arr.shape[0]
    return as_strided(arr,
        shape=(n - window_size + 1, window_size) + arr.shape[1:],
        strides=(arr.strides[0] * step,) + arr.strides)

window_size = 80
step = 1

sliding_data = sliding_window_view(data, window_size, step)
```

**Figure 8:** Example of Python code for processing multiple Numpy files and creating a sliding window view of the data.

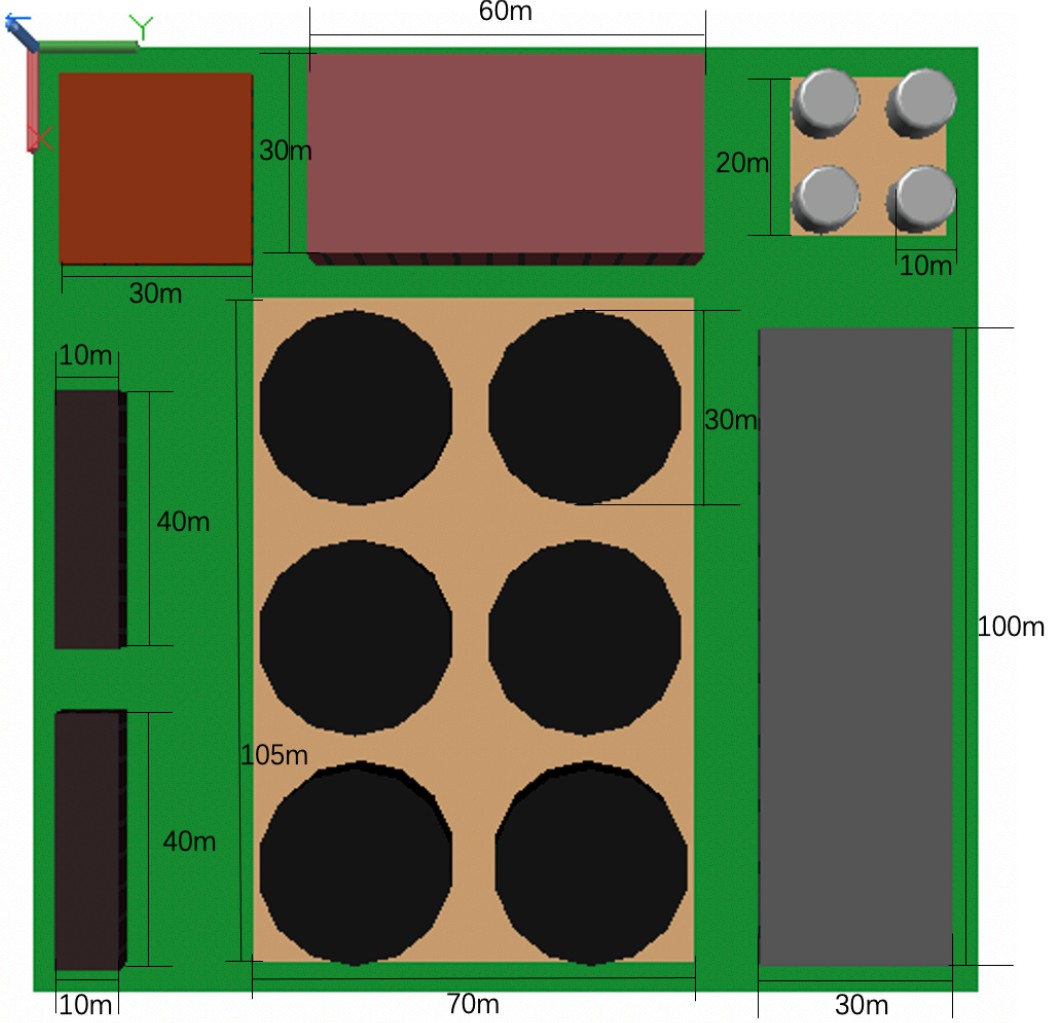

**Figure 9:** top-view

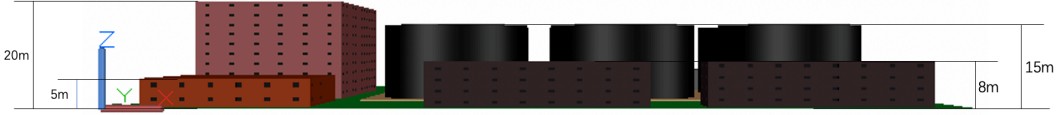

**Figure 10:** side-view

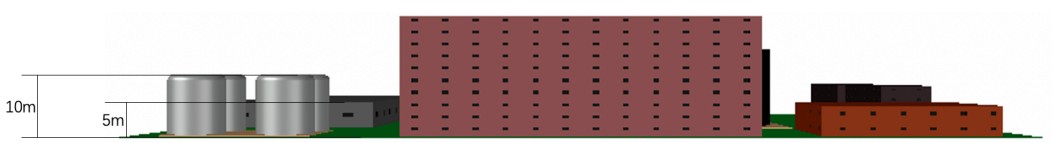

**Figure 11:** back-view

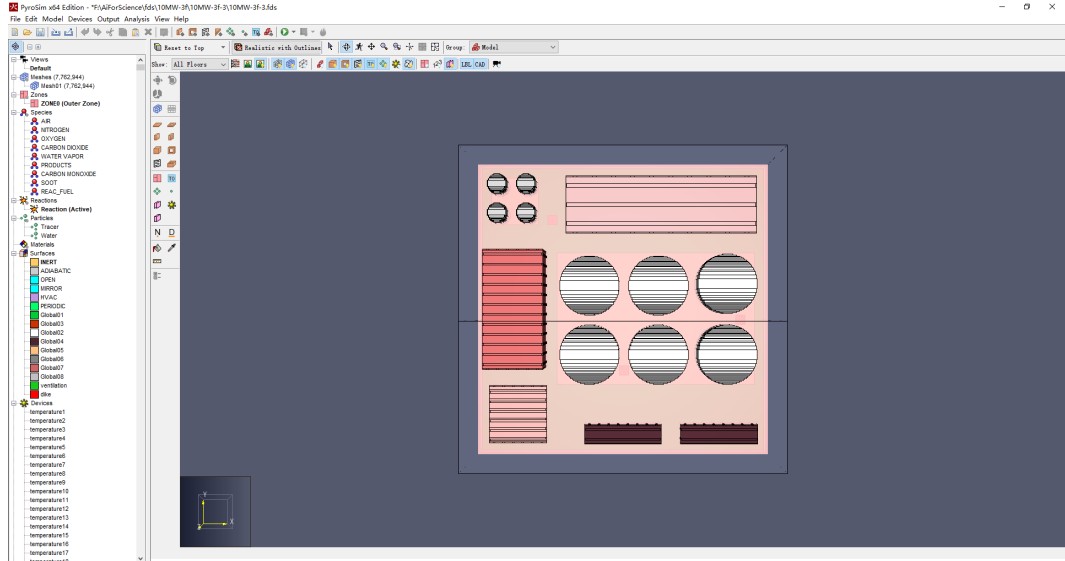

**Figure 12:** The figure shows the design interface of Pyrosim software, displaying the factory scenario used in this study.

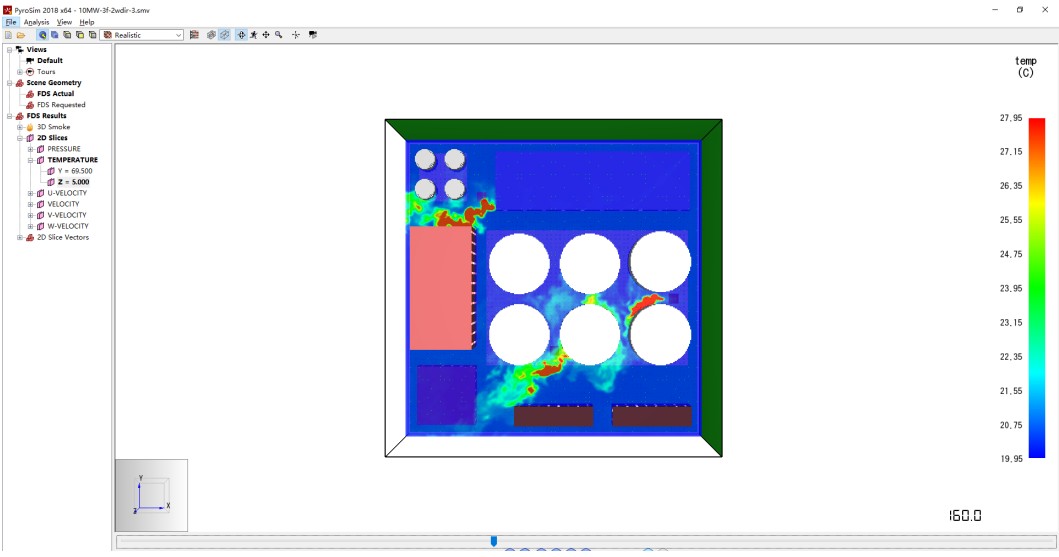

**Figure 13:** This figure demonstrates the use of the PyrosimSimResult feature in the Pyrosim software to visualize the results of numerical simulations.

```python
def merge_fds_files(original_file, output_file):
    # Read the original file content
    with open(original_file, 'r') as f:
        original_content = f.readlines()

    # Find the index of the line containing &SURF ID='fire'
    surf_index = None
    for i, line in enumerate(original_content):
        if "&SURF ID='fire'," in line:
            surf_index = i
            break

    if surf_index is None:
        print("Line containing &SURF ID='fire' not found")
        return

    # Save the new parameters
    new_content = []
    # Sensor spacing
    dist = 1
    startx = 0.0
    starty = 0.0
    # Sensors fixed at a height of 5m plane
    z = 5

    # Add temperature field sensors
    sum = 0
    for i in range(300):
        for j in range(300):
            x = round(startx + j * dist, 1)
            y = round(starty + i * dist, 1)
            sum += 1
            new_content.append("&DEVC ID=" + f"'temperature{sum}', " + "
QUANTITY='TEMPERATURE', XYZ=" + str(x) + "," + str(y) + "," + str(z)
+ "/" + "\n")
    new_content.append("\n")

    # Add velocity field sensors
    ......

    # Add pressure sensors
    ......

    # Insert the new content at the specified position in the original
    content
    merged_content = original_content[:surf_index] + new_content +
original_content[surf_index:]

    # Write the merged content to a new file
    with open(output_file, 'w') as f:
        f.writelines(merged_content)

    print("Merge completed, result saved to", output_file)
```

**Figure 14:** Pseudocode for processing an fds file and adding sensor data in Python.

```python
import numpy as np
import pandas as pd

# CSV file paths
csv_file_path = ['10MW-3f-2wdir-1_devc.csv','10MW-3f-2wdir-2_devc.csv','
    10MW-3f-2wdir-3_devc.csv','10MW-3f-2wdir-4_devc.csv','10MW-3f-2wdir-5
    _devc.csv']
# Paths to save the npy files
npy_file_path = ['10MW-3f-2d-400x3x300x300-1.npy','10MW-3f-2d-400
    x3x300x300-2.npy','10MW-3f-2d-400x3x300x300-3.npy','10MW-3f-2d-400
    x3x300x300-4.npy', '10MW-3f-2d-400x3x300x300-5.npy']

# Clean and filter each CSV file
for i in range(len(csv_file_path)):
    # Read the CSV file
    df = pd.read_csv(csv_file_path[i])

    # Drop the first column
    df = df.drop(df.columns[0], axis=1)

    # Drop the first two rows
    df = df.iloc[2:]   # iloc is used for row-based slicing

    # Convert all data to float type
    df = df.astype(float)

    # Convert the DataFrame to a NumPy array
    data = df.values

    # Define the length of each sub-column
    sub_column_length = 300

    # Current dimensions are 400x27000 (T*dnum)
    # Target dimensions are (T, dtype, row, col)
    # Reshape data to (400, 3, 300, 300)
    reshaped_data = data.reshape(400, 3, sub_column_length, -1)

    print(reshaped_data.shape)

    # Save as .npy file
    np.save(npy_file_path[i], reshaped_data)
```

**Figure 15:** Pseudocode for filtering and cleaning a CSV file, transforming its dimensions, and finally saving it as a .npy file in Python.

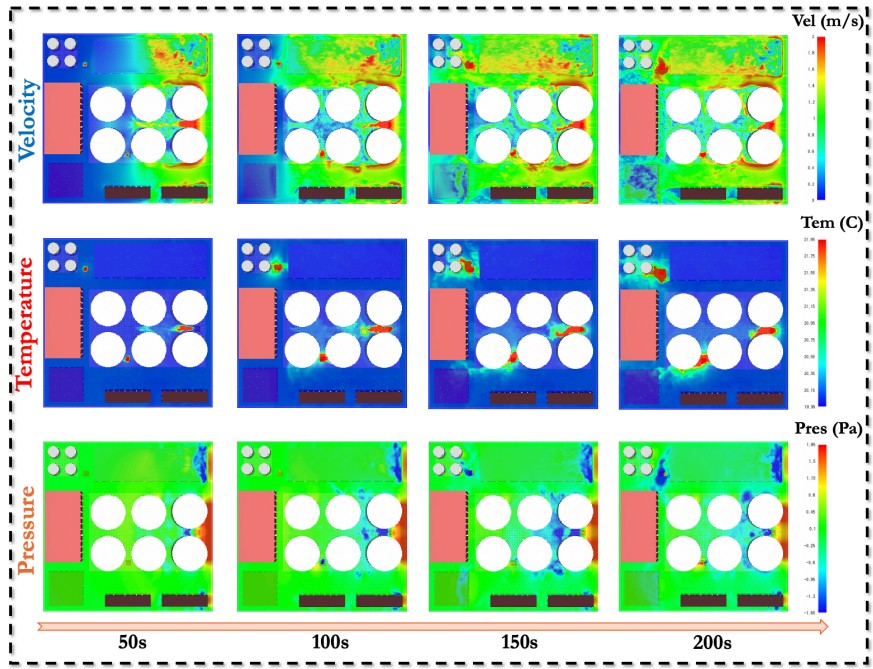

**Figure 16:** The figure illustrates the evolution of the fire under the conditions of a heat release rate of 10 MW, three ignition sources, and a single wind direction with a wind speed of 1 m/s. The fire growth coefficient is set to 0.178. Each frame in the sequence is separated by an interval of 50 seconds.

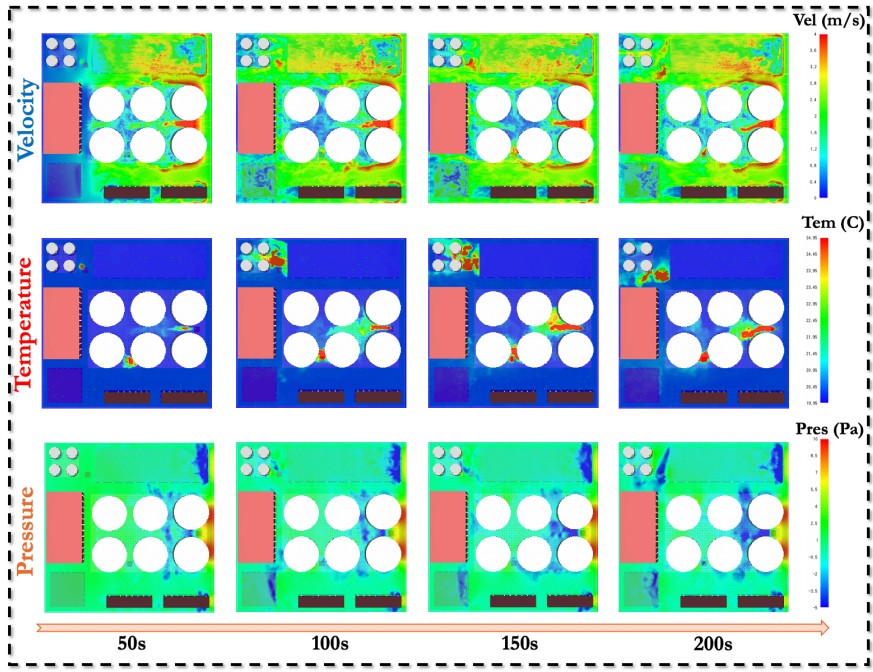

**Figure 17:** The figure illustrates the evolution of the fire under the conditions of a heat release rate of 10 MW, three ignition sources, and a single wind direction with a wind speed of 2 m/s. The fire growth coefficient is set to 0.178. Each frame in the sequence is separated by an interval of 50 seconds.

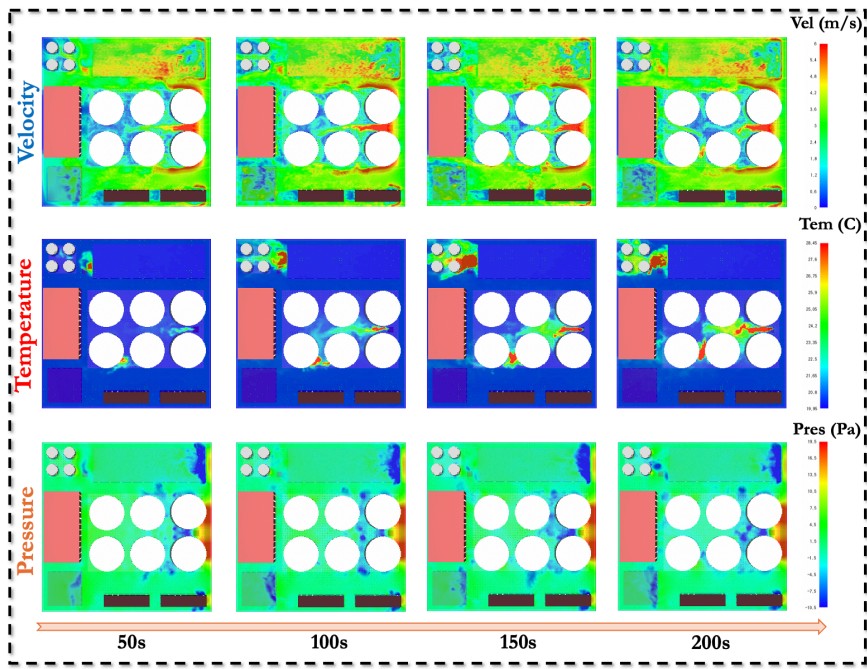

**Figure 18:** The figure illustrates the evolution of the fire under the conditions of a heat release rate of 10 MW, three ignition sources, and a single wind direction with a wind speed of 3 m/s. The fire growth coefficient is set to 0.178. Each frame in the sequence is separated by an interval of 50 seconds.

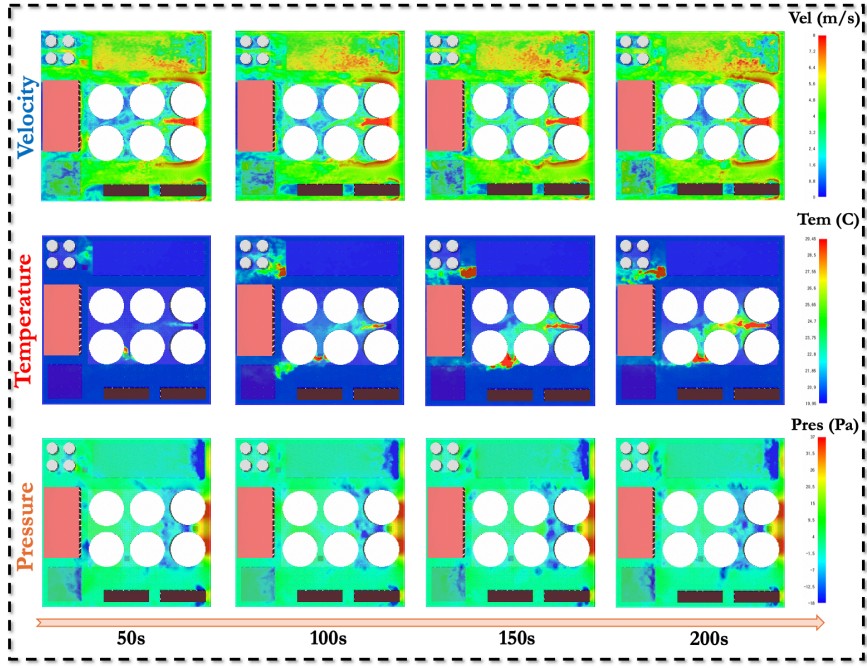

**Figure 19:** The figure illustrates the evolution of the fire under the conditions of a heat release rate of 10 MW, three ignition sources, and a single wind direction with a wind speed of 4 m/s. The fire growth coefficient is set to 0.178. Each frame in the sequence is separated by an interval of 50 seconds.

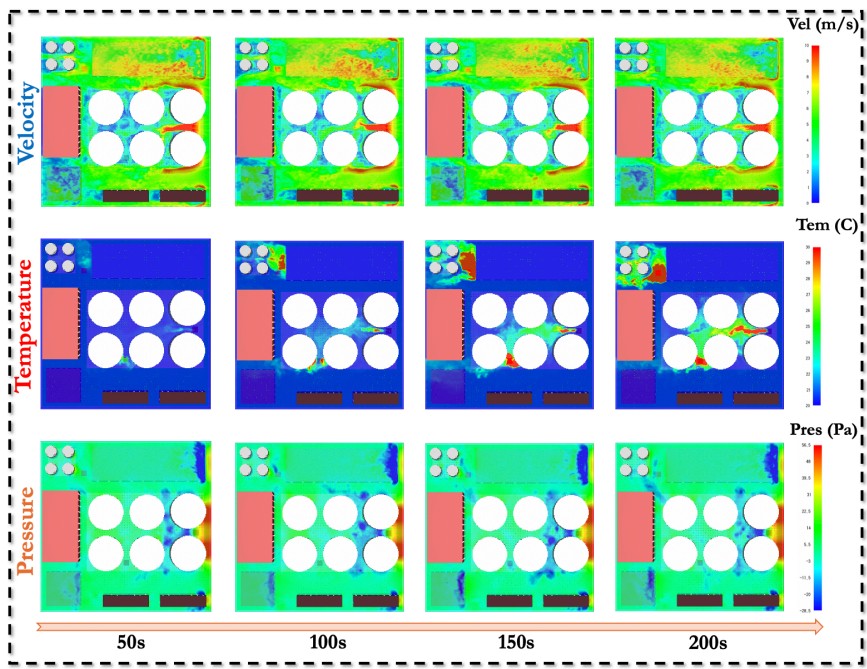

**Figure 20:** The figure illustrates the evolution of the fire under the conditions of a heat release rate of 10 MW, three ignition sources, and a single wind direction with a wind speed of 5 m/s. The fire growth coefficient is set to 0.178. Each frame in the sequence is separated by an interval of 50 seconds.

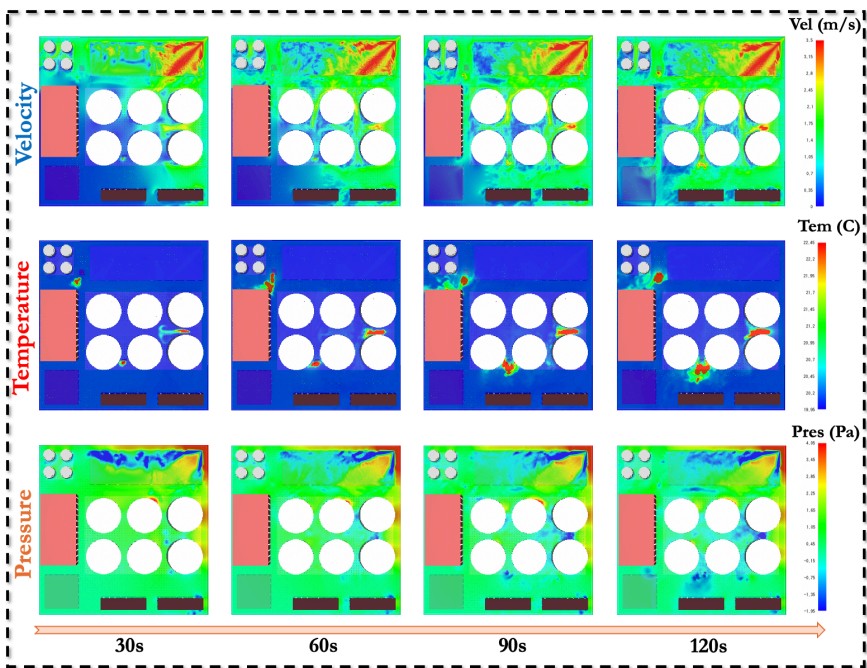

**Figure 21:** This figure shows the evolution of the fire under the conditions of a heat release rate of 10 MW, three ignition sources, and two wind directions with a wind speed of 1 m/s. The fire growth coefficient is set to 0.178. Each frame in the sequence is separated by an interval of 30 seconds.

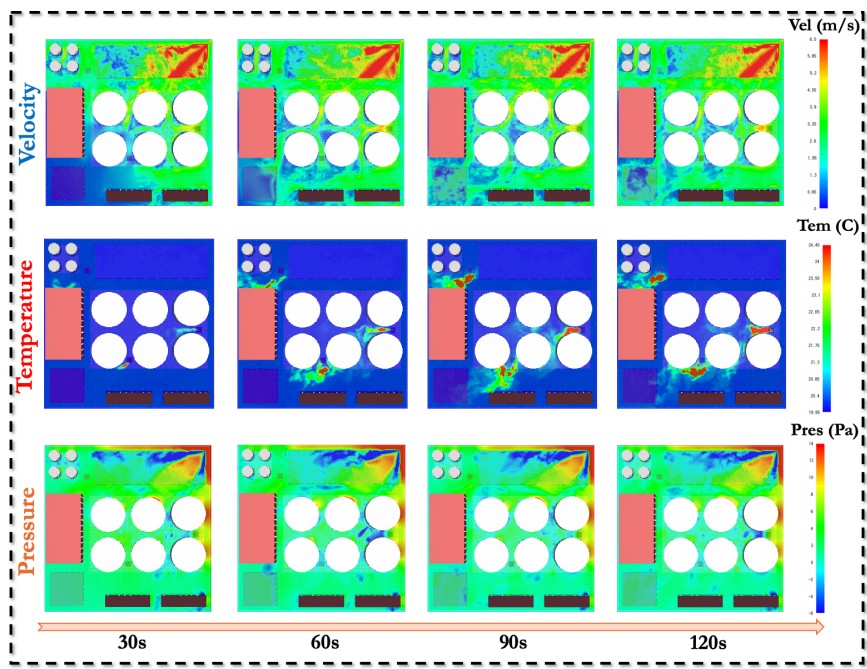

**Figure 22:** This figure shows the evolution of the fire under the conditions of a heat release rate of 10 MW, three ignition sources, and two wind directions with a wind speed of 2 m/s. The fire growth coefficient is set to 0.178. Each frame in the sequence is separated by an interval of 30 seconds.

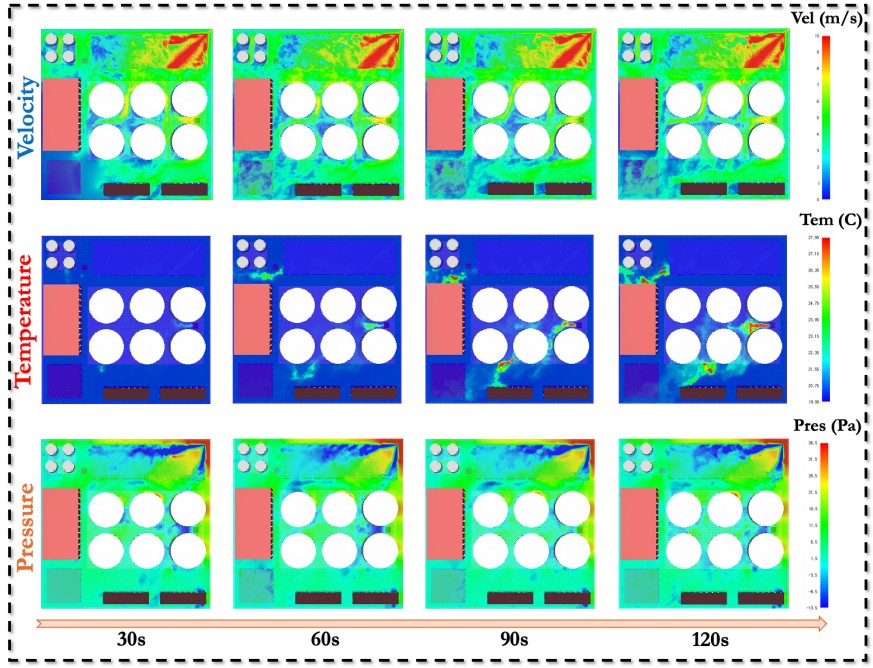

**Figure 23:** 1This figure shows the evolution of the fire under the conditions of a heat release rate of 10 MW, three ignition sources, and two wind directions with a wind speed of 3 m/s. The fire growth coefficient is set to 0.178. Each frame in the sequence is separated by an interval of 30 seconds.

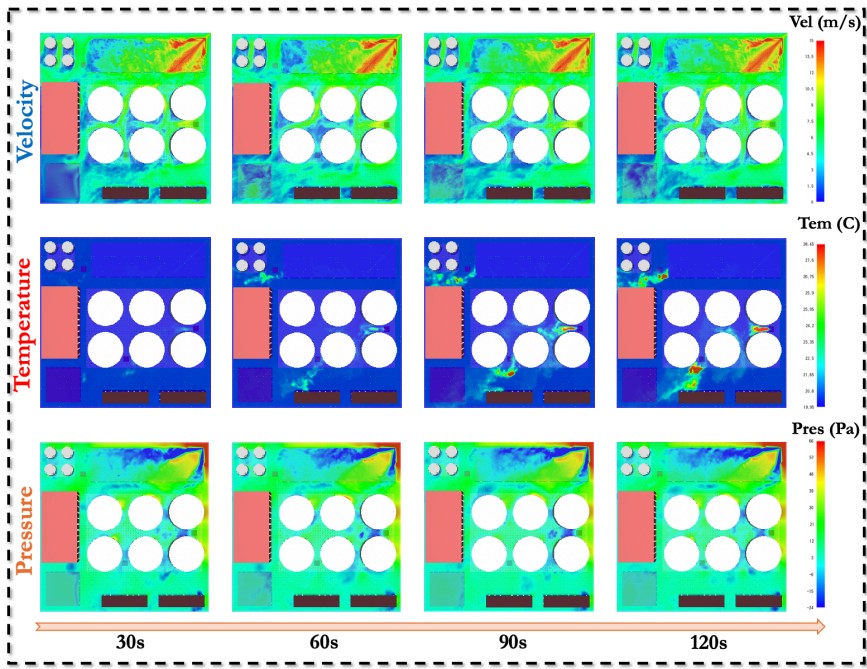

**Figure 24:** This figure shows the evolution of the fire under the conditions of a heat release rate of 10 MW, three ignition sources, and two wind directions with a wind speed of 4 m/s. The fire growth coefficient is set to 0.178. Each frame in the sequence is separated by an interval of 30 seconds.

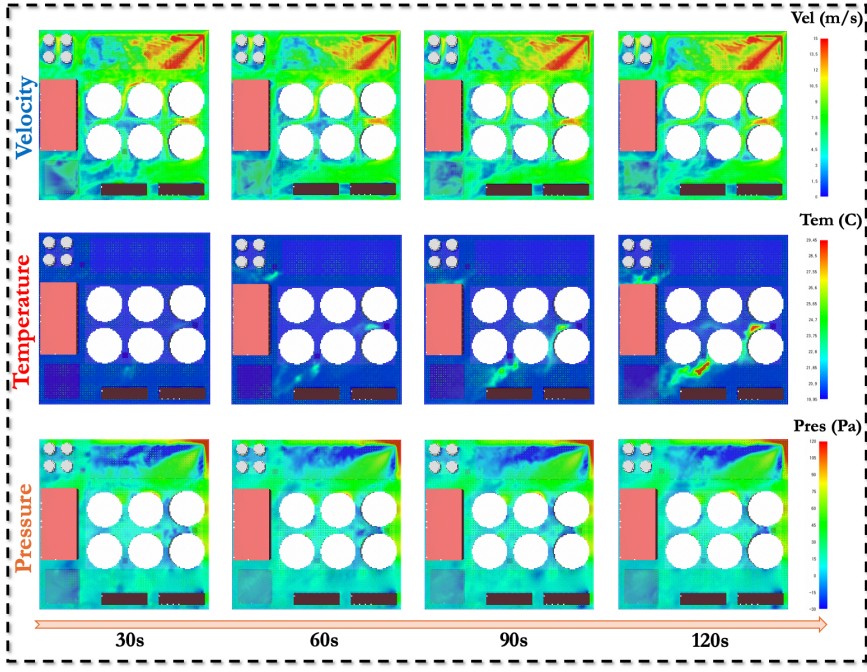

**Figure 25:** This figure shows the evolution of the fire under the conditions of a heat release rate of 10 MW, three ignition sources, and two wind directions with a wind speed of 5 m/s. The fire growth coefficient is set to 0.178. Each frame in the sequence is separated by an interval of 30 seconds.

**Table 4:** Baseline Model Configurations. This table summarizes the configurations of various baseline models, including their hidden dimension, number of layers, learning rate, and dropout rate. These models are used to handle spatiotemporal sequence data efficiently, with different configurations affecting their complexity, convergence speed, and overfitting prevention.

| Model | Hidden Dim | Layers | Learning Rate | Dropout Rate |
|---|---|---|---|---|
| ConvLSTM | 128 | 3 | 0.001 | 0.2 |
| PredRNN | 256 | 4 | 0.001 | 0.3 |
| PhyDNet | 128 | 5 | 0.0005 | 0.2 |
| MIM | 256 | 4 | 0.001 | 0.2 |
| PredRNNv2 | 256 | 6 | 0.0005 | 0.3 |
| PastNet | 128 | 3 | 0.001 | 0.2 |
| SimVP | 128 | 3 | 0.001 | 0.2 |
| SimVPv2 | 128 | 4 | 0.0005 | 0.2 |
| Earthfarseer | 256 | 5 | 0.0005 | 0.3 |
| ResNet | 256 | 6 | 0.0001 | 0.2 |
| U-Net | 128 | 5 | 0.001 | 0.3 |
| ViT | 256 | 6 | 0.0001 | 0.2 |
| Swin Transformer | 256 | 6 | 0.0001 | 0.2 |
| MLP-Mixer | 256 | 4 | 0.0005 | 0.2 |
| FNO | 128 | 3 | 0.001 | 0.2 |
| LSM | 128 | 3 | 0.001 | 0.2 |
| CNO | 128 | 4 | 0.001 | 0.2 |
| UNO | 256 | 5 | 0.0005 | 0.2 |
| NMO | 256 | 5 | 0.0005 | 0.2 |

## B  BASELINE DESCRIPTIONS

- **ConvLSTM** Shi et al. (2015): Combines convolutional neural networks and LSTM for spatiotemporal sequence prediction, suitable for handling spatiotemporal data.

- **PredRNN** Wang et al. (2017): Uses a recurrent neural network with a multi-level recurrent structure and cross-memory cells to improve spatiotemporal feature modeling.

- **PhyDNet** Guen & Thome (2020): Combines explicit physical constraints with implicit neural network models for video prediction, enhancing prediction accuracy.

- **MIM** Wang et al. (2019): Utilizes a memory-in-memory mechanism for video prediction, capturing complex spatiotemporal patterns.

- **PredRNNv2** Wang et al. (2022b): An improved version of PredRNN with a deeper recurrent structure and more complex memory units for better spatiotemporal prediction performance.

- **PastNet** Wu et al. (2023c): Uses past time step data for prediction, with an efficient spatiotemporal prediction architecture achieving good performance with fewer parameters.

- **SimVP** Gao et al. (2022a): A spatiotemporal prediction model based on simple visual transformations, featuring high computational efficiency.

- **SimVPv2** Tan et al. (2022): An improved version of SimVP with optimized visual modules and additional feature transformation layers, enhancing prediction performance.

- **Earthfarseer** Wu et al. (2024a): Designed for earth system prediction, combining various spatiotemporal feature extraction methods to improve accuracy in earth science predictions.

- **ResNet** He et al. (2016): A deep residual network that addresses the vanishing gradient problem in deep networks, widely used in image recognition and classification tasks.

- **U-Net** Ronneberger et al. (2015): A convolutional neural network-based image segmentation model with an encoder-decoder structure, widely used in medical image processing and spatiotemporal prediction tasks.

- **ViT** Dosovitskiy et al. (2020): A Vision Transformer model that captures long-range dependencies in images through the self-attention mechanism.

- **Swin Transformer** Liu et al. (2021): An improved Transformer model with a hierarchical window attention mechanism for processing high-resolution images.
- **MLP-Mixer** Tolstikhin et al. (2021): A fully connected neural network that performs image classification by mixing features and positional encodings, simplifying the design of traditional convolutional networks.
- **FNO** Li et al. (2020): A Fourier Neural Operator model for solving partial differential equations using Fourier transforms to improve computational efficiency.
- **LSM** Wu et al. (2023a): Uses state-space models for spatiotemporal prediction, learning state transition and observation equations for high-precision forecasting.
- **CNO** Raonic et al. (2024): Combines convolutional neural networks and neural operator models to capture local spatiotemporal features for efficient PDE solving.
- **UNO** Ashiqur Rahman et al. (2022): A U-shaped memory enhanced architecture that allows for deeper neural operators.
- **NMO** Wu et al. (2024d): An operator learning paradigm for learning the intrinsic dimension representation of the underlying operator.

## C BASELINE MODEL CONFIGURATIONS

This section describes the baseline models and their configurations, including model name, hidden dimension, number of layers, learning rate, and dropout rate. Table 4 summarizes the configurations for each model.

The models include traditional convolutional neural networks, recurrent neural networks, and recent Transformer models. These models handle spatiotemporal sequence data efficiently. The hidden dimension and number of layers determine model complexity, the learning rate affects convergence speed, and the dropout rate helps prevent overfitting. Comparing these baseline models' performance provides insights into their effectiveness for specific tasks and guides further research.

## D LIMITATIONS AND FUTURE WORK

### D.1 LIMITATIONS

- **Scenario Limitation**: Although the dataset considers various fire scenarios and environmental variables, real-world fires are often more complex, with greater diversity and unpredictable variations. For example, differences in structural features, building materials, and crowd density may not be fully represented in the dataset.
- **Model Generalization**: Since the dataset is generated under controlled conditions, the model may struggle to handle the complex and dynamic fire scenarios encountered in the real world. Actual fires may involve different fire sources, combustible materials, and changing building structures, which could fall outside the scope of the current dataset.
- **Environmental Factors**: Real-world fire development is influenced by many uncontrollable factors, such as climate conditions, weather changes, and evacuation situations. These factors might not have been fully considered in the simulation, leading to reduced prediction accuracy when the model is applied to actual fire scenarios.

### D.2 FUTURE WORK

To adapt a model pre-trained on the Open-CK dataset for real-world fire dynamics modeling, the following measures can be taken:

- **Data Augmentation**: Introduce more real-world fire scenario data to enhance the model's generalization ability. For example, by incorporating actual fire records and incident data, the deficiencies of the Open-CK dataset can be addressed, adding diversity to the scenarios.
- **Transfer Learning**: Pre-train the model on the Open-CK dataset and then fine-tune it on a fire dataset that more closely resembles real-world conditions. This approach allows

the model to retain the fundamental fire dynamics features learned from Open-CK while adapting to new environments and scenarios.

