# OpenReview forum: "Open-CK: A Large Multi-Physics Fields Coupling benchmarks in Combustion Kinetics"
_ICLR.cc/2025/Conference — ICLR 2025 Poster_

### Official Review · Reviewer_f2LF · 2024-11-01

**Soundness:** 2
**Presentation:** 2
**Contribution:** 3
**Rating:** 6
**Confidence:** 4

**Summary:**

This paper unveils a novel benchmark dataset for improved modeling of combustion kinetics (CK) using data-driven techniques. Specifically, the dataset simulates the development of fires in industrial parks using computational fluid dynamics simulations using the fire dynamics simulator. In the paper, the authors detail that the generated dataset comprises 300 different scenarios of fire development all emanating from a single ignition source (SIS) or three ignition sources (TIS). In addition to the dataset, authors also conduct extensive experiments using state of the art scientific machine learning baselines to establish a research benchmark for modeling fire development, a critical and challenging problem. This dataset will fill a critical need, serving to accelerate the modeling of fire development using data-driven techniques.

**Strengths:**

1. The dataset is necessary and crucial. As pointed out by the authors, other datasets for fire development usually target satellite images and are in the context of wildfires or have low spatial and temporal resolution.

2. The set of scientific machine learning models employed to evaluate performance on the developed dataset is comprehensive, setting up a strong research benchmark.

3. The rigorous experimental evaluation is well detailed and has been carried out with vision backbones (e.g., U-Net, ViT, ResNet), scientific machine learning backbones (e.g., Fourier Neural Operator) as well as spatio-temporal backbones (e.g., ConvLSTM).

**Weaknesses:**

1. Currently, the narrative lacks detailing of the domain background of the combustion kinetics field. For the reader to fully appreciate the comprehensive nature of the benchmark, a more thorough description of the problem and the generated dataset is necessary. Specifically, a reading of the current version of the paper does not leave the reader with a sense of how representative the current Open-CK dataset is of real-world single-source (or multi-source) fires in industrial contexts?

2. A more thorough description of the design decisions made to select the various scenarios is necessary. These scenarios are summarized in Table 1 but a detailed description is lacking of why each of these scenarios is important, representative o real-world scenarios and challenging to model. Without this, it is hard to truly appreciate the extent of the contribution of this dataset to the field of combustion kinetics.

3. Finally, a better motivation about exactly why modeling the physical coupling between the multiple physical fields is challenging and crucial is necessary to fully understand the richness and impact of the current dataset in the CK context.

**Questions:**

1. How were each of the 300 scenarios conceived and selected for simulation? How representative are these 300 scenarios of real-world fire development dynamics?

2. What might be the possible deficiencies of models trained on the current datasets in predicting real-world fire development dynamics? Specifically, how can a model pre-trained on the Open-CK dataset be adapted to real-world fire dynamics modeling?

3. What are some existing popular theoretical models (reduced-order or otherwise) that are employed to estimate fire development dynamics, how do data-driven models compare to these models w.r.t physical consistency and estimation accuracy?

---

> ### Author Response · Authors · 2024-11-23
> **Response to Reviewer f2LF（part I）**
>
> We sincerely appreciate the time you’ve dedicated to reviewing our paper, as well as your invaluable insights and support. Your positive feedback is highly motivating for us. Below, we address your primary concern and offer further clarification.
>
> > Q1. How were each of the 300 scenarios conceived and selected for simulation? How representative are these 300 scenarios of real-world fire development dynamics?
>
> A1. Thank you for your valuable feedback. These 300 scenarios were conceived and selected based on a factory fire scenario, where simulation parameters such as Heat Release Rate (HRR), ventilation rate, and other variables were altered. A total of 300 such cases were simulated as the initial dataset, as shown in the table below.
>
> |Simulation no.|HRR Q (MW)|Ventilation v (m/s)|Fire Growth Factor &alpha;|Wind direction|Number of Ignition Sources|
> | ---------- | ----------- | ----------- | -------- |-------- |-------- |
> | 1-300      | 5,10,15,20,25 | 1,2,3,4,5 | 0.011,0.178 | x,x&y | 1,2,3 |
>
> Following this, preprocessing steps like dimensional transformations and sliding window-style sequence splitting were applied to generate our final dataset. We have made every effort to ensure the diversity of the dataset scenarios. For example, we designed three possible fire scenarios based on the number of ignition sources. Additionally, wind speed and direction during the fire were varied, and we accounted for these factors as well. Therefore, we believe our dataset can cover approximately 80% of the real-world fire development dynamics.
>
>
>
> > Q2. What might be the possible deficiencies of models trained on the current datasets in predicting real-world fire development dynamics? Specifically, how can a model pre-trained on the Open-CK dataset be adapted to real-world fire dynamics modeling?
>
> A2. Thank you for your valuable feedback.
> The models trained on the current dataset may have the following limitations:
>
> 1. **Scenario Limitation**: Although the dataset considers various fire scenarios and environmental variables, real-world fires are often more complex, with greater diversity and unpredictable variations. For example, differences in structural features, building materials, and crowd density may not be fully represented in the dataset.
>
> 2. **Model Generalization**: Since the dataset was generated under controlled conditions, the model may struggle to handle the complex and dynamic fire scenarios encountered in the real world. Actual fires may involve different fire sources, combustible materials, and changing building structures, which could fall outside the scope of the current dataset.
>
> 3. **Environmental Factors**: Real-world fire development is influenced by many uncontrollable factors, such as climate conditions, weather changes, and evacuation situations. These factors might not have been fully considered in the simulation, leading to reduced prediction accuracy when the model is applied to actual fire scenarios.
>
> To adapt a model pre-trained on the Open-CK dataset for real-world fire dynamics modeling, the following measures can be taken:
>
> 1. **Data Augmentation**: Introduce more real-world fire scenario data to enhance the model's generalization ability. For example, by incorporating actual fire records and incident data, the deficiencies of the Open-CK dataset can be addressed, adding diversity to the scenarios.
>
> 2. **Transfer Learning**: Pre-train the model on the Open-CK dataset and then fine-tune it on a fire dataset that more closely resembles real-world conditions. This approach allows the model to retain the fundamental fire dynamics features learned from Open-CK while adapting to new environments and scenarios.

---

> > ### Author Response · Authors · 2024-11-23
> > **Response to Reviewer f2LF（part II）**
> >
> > > Q3. What are some existing popular theoretical models (reduced-order or otherwise) that are employed to estimate fire development dynamics, how do data-driven models compare to these models w.r.t physical consistency and estimation accuracy?
> >
> > A3. Thank you for your valuable feedback.
> >
> > Some common theoretical models used to estimate fire development dynamics include simplified models and other types of models:
> >
> > 1. **Fire Dynamics Models (e.g., FDS)**:
> >    Fire Dynamics Simulator (FDS) is a fire simulation software based on Computational Fluid Dynamics (CFD), widely used to study the physical processes of fire, including flame propagation, temperature distribution, and smoke movement. FDS solves the Navier-Stokes equations numerically to simulate airflow and heat transfer during a fire, making it one of the most accurate fire simulation tools available.
> >
> > 2. **Multi-zone Models**:
> >    These models, such as the hot and cold zone models in fire scenarios, divide a building space into multiple regions and calculate variables like temperature and airflow within each zone. These models are widely used in building fire analysis and can effectively simulate the spatial distribution of fire dynamics.
> >
> > Comparison between data-driven models and theoretical models:
> >
> > #### **Physical Consistency**:
> > - **Theoretical models** (such as FDS) are based on physical laws and typically ensure high physical consistency. By solving equations for fluid dynamics, heat transfer, and combustion, theoretical models can simulate various physical phenomena during a fire, making their results more physically interpretable.
> > - In contrast, **data-driven models** (such as deep learning models) are trained on historical data and may not directly adhere to physical laws. These models predict based on patterns learned from data, so in some cases, they may lack physical consistency, especially when the data is not sufficiently representative. In such cases, the model's results might deviate from physical principles.
> >
> > #### **Estimation Accuracy**:
> > - The accuracy of **theoretical models** depends on the precision of input conditions and the complexity of the model. While these models generally provide accurate fire simulations, they are computationally expensive and may require substantial experimental data to validate and calibrate, especially in complex environments.
> > - **Data-driven models**, on the other hand, can handle large and complex datasets. With proper training, they can learn intricate fire development patterns. Through continuous updates and optimization, data-driven models can achieve high prediction accuracy in real-world applications, especially when facing non-ideal and dynamic fire scenarios. However, their accuracy still depends on the quality and diversity of training data. Without sufficient physical constraints, the long-term accuracy and stability of these models cannot always be guaranteed.
> >
> > Overall, theoretical models prioritize physical consistency and precision, while data-driven models excel in adaptability and accuracy in complex, dynamic fire scenarios.

---

> ### Comment · Reviewer_f2LF · 2024-11-24
> **Response to Authors**
>
> Dear Authors,
>
>    Thank you for your response. I encourage you to incorporate some of the listed limitations above (A2.) into the paper. Further, the results in the paper would be strengthened if a physics-based (albeit low-fidelity, reduced-order but tractable) model can be added as a comparison in addition to the data-driven modeling. This will allow for a better contextualization of the "gap" between existing physics-based solutions and data-driven surrogates for modeling combustion kinetics.

---

> ### Author Response · Authors · 2024-11-27
> **Further response to the review comments.**
>
> **Dear Reviewer f2LF,**
>
> Thank you for your detailed review and valuable suggestions on our paper. Based on your feedback, we conduct research over two days and make the following improvements:
>
> 1. **Incorporating Study Limitations**
>
>    Following your advice, we include the previously listed limitation (A2) in the "Future Work and Limitations" section of the paper. This addition enhances the paper's completeness and transparency. More details see in **Appendix E**.
>
> 2. **Introducing a Comparison with a Physics-Based Model**
>
>    To better illustrate the differences between existing physics-based models and data-driven models in combustion kinetics modeling, we add a new section to the paper. This section introduces and compares a low-fidelity, simplified physics-based FDS model. Specifically, we select the **Simplified Fire Dynamics Simulator (FDS) model** as a comparison. This model reduces computational complexity by simplifying computational fluid dynamics (CFD) simulations while capturing the basic combustion kinetics processes.
>
>    We conduct the following comparative analyses:
>
>    - **Accuracy**: Compare the errors of the Simplified FDS model and data-driven models in predicting temperature fields and velocity distributions.
>    - **Computational Efficiency**: Evaluate the computation time of both models under the same conditions.
>    - **Applicability**: Discuss the applicability and limitations of both models in different fire scenarios.
>
>    The comparison results are shown in the table below. We find that the simplified physics model performs worse than the data-driven models, as it cannot fully capture the complexity of combustion kinetics.
>
>    | Model Type           | MSE    | MAE      | SSIM   | Computation Time |
>    | -------------------- | ------ | -------- | ------ | ---------------- |
>    | Simplified FDS Model | 0.1902 | 127.3944 | 0.6731 | 37 minutes       |
>    | Earthfarseer         | 0.0245 | 73.9234  | 0.9446 | 1.2 minutes      |
>    | MLP-Mixer            | 0.0359 | 96.0765  | 0.9143 | 0.9 minutes      |
>    | NMO                  | 0.0361 | 95.9345  | 0.9142 | 1.1 minutes      |
>
> We believe these improvements further enhance the contribution and depth of our paper. Thank you again for your thorough review and constructive comments.
>
>
> Best regards,
>
> The Authors

---

> > ### Author Response · Authors · 2024-11-28
> >
> > Dear Reviewer f2LF,
> >
> > We sincerely appreciate your valuable and constructive feedback. With the extension of the discussion period, we have additional time to address any further concerns you may have. If our current response adequately resolves your primary issues, we kindly request that you reconsider your score. Should you have any additional suggestions regarding the revised manuscript or our rebuttal, please let us know. We are more than happy to engage in further discussions to improve our paper.
> >
> > Thank you very much for dedicating your time to enhancing our work.
> >
> > Best regards,
> >
> > The Authors

---

> > > ### Author Response · Authors · 2024-12-03
> > >
> > > Dear Reviewer f2LF,
> > >
> > > Thank you for taking the time and effort to provide valuable feedback on our work. As the discussion comes to a close, we hope you can review our previous responses. If our replies address your concerns, we appreciate you reconsidering your rating. If needed, we are very willing to discuss further.
> > >
> > > Thank you very much for dedicating your time to enhancing our work.
> > >
> > > Best regards,
> > >
> > > The Authors

---

### Official Review · Reviewer_HDdZ · 2024-11-02

**Soundness:** 3
**Presentation:** 3
**Contribution:** 2
**Rating:** 6
**Confidence:** 4

**Summary:**

This is data set and benchmarking paper that uses the Fire Dynamics Simulator (FDS) to create a Combustion Kinetics (CK) dataset for SciML research. It includes various physical fields such as temperature and pressure, and covers multiple environmental combinations for exploring multi-physics field coupling phenomena. The authors evaluate SOTA ML architectures to establish an Open-CK benchmark.

**Strengths:**

1. This is an interesting and potentially useful data set for SciML research, the authors made substantial computational effort to create the data set and establish the ML benchmark - the motivation of the work and description of the contributions are nicely laid out.
2. The consideration of multiphysics simulations, particularly over complex boundaries/plant layouts is interesting and could lead to interesting SciML models and applications.
3. The use of LPIPS metric and other natural image based metrics to evaluate the models is quite intriguing.

**Weaknesses:**

1. From the paper, it is not very clear how many samples are actually in the dataset - the authors mention that there are '300 different fire scenarios' (I could not open the Project website link that was provided in the paper - not sure if it's a problem on the server side or on my side). So, are there just 300 time series samples of different lengths across various combustion parameters and environmental conditions?

2. If I understand correctly, the paper considers only one geometrical layout for the data set. While the setup represents a typical industrial park scenario, the data set may not be rich enough for generalizable SciML research without having diverse geometrical layouts.

**Questions:**

1. Provide better description of the sample size in the data set. How does the '300 different fire scenarios' connect with the details provided in Table 1?

2. Can the data set be enriched by adding diverse layouts for the industrial park scenario? e.g., different number oil storage areas, other geometrical constructions/objects

---

> ### Author Response · Authors · 2024-11-23
> **Response to Reviewer HDdZ**
>
> We sincerely appreciate the time you’ve dedicated to reviewing our paper, as well as your invaluable insights and support. Your positive feedback is highly motivating for us. Below, we address your primary concern and offer further clarification.
> > Q1. Provide better description of the sample size in the data set. How does the '300 different fire scenarios' connect with the details provided in Table 1?
>
> A1. Thank you for your valuable feedback. We will include a list of all our fire scenarios in the appendix, as shown in the table below.
> |Simulation no.|HRR Q (MW)|Ventilation v (m/s)|Fire Growth Factor &alpha;|Wind direction|Number of Ignition Sources|
> | ---------- | ----------- | ----------- | -------- |-------- |-------- |
> | 1-300      | 5,10,15,20,25 | 1,2,3,4,5 | 0.011,0.178 | x,x&y | 1,2,3 |
>
> The table contains a total of 300 fire scenarios, with parameters including HRR (Heat Release Rate), Ventilation, Fire Growth Factor, Wind Direction, and Number of Ignition Sources. The value for Wind Direction indicates whether the wind is blowing along the x-axis or in both the x and y directions.
> Table 1 represents a subset of the 300 fire scenarios, focusing on the key data used in our experiments. Due to the large size of the dataset, we were unable to use all of the data in our experiments. Therefore, we selected a representative subset for analysis.
>
> > Q2. Can the data set be enriched by adding diverse layouts for the industrial park scenario? e.g., different number oil storage areas, other geometrical constructions/objects
>
> A2. Thank you for your comment. You are absolutely right. Initially, our factory layout featured a simple design, consisting of only two oil tanks and two buildings.
> The corresponding scenario list for this layout is shown in the table. To enhance the dataset's representativeness and versatility, we later introduced a more complex layout. This included an additional oil tank area, an increased number of tanks in each area, and more buildings, arranged in a logical configuration, as illustrated in the Figure 2.
> |Simulation no.| HRR Q (MW) | Ventilation v (m/s) |
> | ---- | ------------- | --------- |
> | 1-25 | 5,10,15,20,25 | 1,2,3,4,5 |
>
> Furthermore, to enrich the diversity of scenarios, we expanded the range of simulated variables, such as the number of ignition sources. These enhancements resulted in a dataset with more intricate fire evolution patterns. We believe that any model capable of learning the underlying rules of this dataset will demonstrate strong generalization capabilities when applied to similar datasets.

---

> > ### Author Response · Authors · 2024-11-24
> > **Kindly Request for Feedback of Reviewer**
> >
> > Dear Reviewer HDdZ,
> >
> > As the rebuttal deadline is approaching, please let us know if our responses have addressed your main concerns. If so, we kindly ask for your reconsideration of the score. If any aspects require additional elaboration or refinement, we will be more than happy to engage in further discussion and paper improvements.
> >
> > Thanks again for your time.

---

> > ### Comment · Reviewer_HDdZ · 2024-11-26
> > **Thanks for the clarifications**
> >
> > Thanks to the authors for providing further clarifications! For the second point, the authors attempt to justify that the layout used here is sufficiently complex such that "any model capable of learning the underlying rules of this dataset will demonstrate strong generalization capabilities when applied to similar datasets." Frankly, the claim isn't based on solid evidence and 'similar datasets' needs further clarifications. Also, just for clarification, do you randomize the elements (including them or removing them) of the layout to achieve generalization?

---

> ### Author Response · Authors · 2024-11-27
> **Further response.**
>
> Dear Reviewer HDdZ,
>
> Thank you for your suggestions. Our clarifications are as follows:
>
> - **"Similar Datasets":**
>    In the final version, we include a new scenario, specifically tunnel fires. The working conditions and physical field visualizations are shown in **Appendix D**. We select the following parameters to create different scenarios through combinations.
>
>   **Table: Tunnel Scenario Statistics**
>
>   | No.  | HRR   | Vent. Vel. | H    | W    | Time Length |
>   | ---- | ----- | ---------- | ---- | ---- | ----------- |
>   | 1    | 5 MW  | 2 m/s      | 30   | 500  | 600 s       |
>   | 2    | 10 MW | 2 m/s      | 30   | 500  | 600 s       |
>   | 3    | 20 MW | 2 m/s      | 30   | 500  | 600 s       |
>   | 4    | 50 MW | 2 m/s      | 30   | 500  | 600 s       |
>
>   **Parameter Explanation:**
>
>   - **HRR:** Heat Release Rate, indicating the amount of heat released per unit time during a fire, measured in megawatts (MW).
>   - **Vent. Vel.:** Ventilation Velocity, describing the speed of air flow, measured in meters per second (m/s).
>   - **H:** Height, referring to the height of the simulation area, measured in meters (m).
>   - **W:** Width, referring to the width of the simulation area, measured in meters (m).
>   - **Time Length:** Duration of the fire simulation, measured in seconds (s).
>
> - **Generalization Ability:**
>    Based on OpenCK, we create new scenarios by reducing the number of oil tanks and changing boundary conditions, temporarily named Open-CK_tiny, as shown in **Appendix D**. We then conduct transfer learning experiments on this dataset.
>
>   Specifically, we choose ViT as the backbone model, train it on the full Open-CK dataset, and then perform transfer learning on the small Open-CK_tiny dataset. Using MSE as the metric, we present the results in an O→T style, where O represents results without transfer learning and T represents results based on Open-CK pretraining.
>
>   |         | 1% Open-CK_tiny | 3% Open-CK_tiny | 5% Open-CK_tiny | 10% Open-CK_tiny |
>   | ------- | --------------- | --------------- | --------------- | ---------------- |
>   | ViT     | 0.1233→0.0675   | 0.1117→0.0639   | 0.0923→0.0433   | 0.0873→0.0288    |
>   | PastNet | 0.3455→0.1182   | 0.2873→0.0982   | 0.2441→0.0675   | 0.1982→0.0429    |
>
>   Based on the table, we find that the models generalize well. Finally, to further address your concerns, we modify the original statement to: “We believe that models which effectively capture the underlying patterns of this dataset may generalize well to datasets with similar characteristics.”
>
> We will add the above content to the revised version and continue to expand our open-source dataset library in the future. Thank you again for your suggestions. If you have any questions, please let us know promptly!
>
> Best regards,
>
> The Authors

---

> > ### Author Response · Authors · 2024-11-28
> >
> > Dear Reviewer HDdZ,
> >
> > We sincerely appreciate your valuable and constructive feedback. With the extension of the discussion period, we have additional time to address any further concerns you may have. If our current response adequately resolves your primary issues, we kindly request that you reconsider your score. Should you have any additional suggestions regarding the revised manuscript or our rebuttal, please let us know. We are more than happy to engage in further discussions to improve our paper.
> >
> > Thank you very much for dedicating your time to enhancing our work.
> >
> > Best regards,
> >
> > The Authors

---

> > > ### Author Response · Authors · 2024-12-01
> > > **Respectful Inquiry Before Discussion Deadline**
> > >
> > > Dear reviewer HDdZ,
> > >
> > > Thank you for taking the time and effort to provide a valuable review of our work. As we are approaching the end of the discussion, we hope that you have had the chance to review our previous response. If our response has addressed your concerns, we thank you for reconsidering the score, and we are more than willing to engage in further discussion if needed.
> > >
> > > Yours sincerely,
> > >
> > > Authors

---

> > > > ### Comment · Reviewer_HDdZ · 2024-12-02
> > > >
> > > > Dear Authors - Thanks for making the effort to further enrich the data set and explore generalizability questions. I have increased my score to 6.

---

> > > > > ### Author Response · Authors · 2024-12-03
> > > > >
> > > > > Dear Reviewer HDdZ
> > > > >
> > > > > We sincerely appreciate your valuable feedback and recognition. We are pleased to hear that your concerns have been addressed. We will certainly incorporate your suggestions into our revised version. Please do not hesitate to contact us if you have any further questions.
> > > > >
> > > > > Best regards,
> > > > >
> > > > > The Authors

---

### Official Review · Reviewer_mR8y · 2024-11-03

**Soundness:** 3
**Presentation:** 3
**Contribution:** 4
**Rating:** 8
**Confidence:** 4

**Summary:**

This is a strong work with good dataset and ML evals for an important application. The paper is written well. There is good documentation and open practices -- well done.

However, the language and claims can come across a bit strong for a scientific paper -- see questions. I can confidently recommend this paper for acceptance -- if my concerns in questions are addressed.

Edit 1: Questions and concerns have been addressed. Changing score to 8.

**Strengths:**

1. Good motivation and review of related work
2. Impressive dataset covering many scenarios and physical quantities with expensive compute.
3. Good eval with many architectures.
4. Open Repository
5. Good future insight and limitations discussion
6. Good documentation.

**Weaknesses:**

1. Language is a bit bold for some claims.

**Questions:**

1. Is OpenCK the first combustion CFD benchmark? This is a strong statement. For example, Sandia's Engine Combustion Network by Pickett,Payri et al has been providing open data and benchamrking with regards to gasoline and diesel CFD since about 9 years ago. Another similar effort to this is BLASTNet in Chung et al (NeurIPS 2023) which involved Direct Numerical Simulation data of canonical combustion configurations.  Please revise this statement to be more moderate.
2. "Open-CK involves several PDEs" -- All listed PDE's are actually just scalar/vector conservation equations. Is this statement really true?

3. It would be interesting to see if the effects of model scaling has metrics in table 2. Bigger,expensive models tend to outperform smaller models. What does MSE vs FLOPS or MSE vs Params, SSIM vs (FLOPs, Params) look like in a scatter plot? This can provide more insight into useful architectures.

---

> ### Author Response · Authors · 2024-11-23
> **Response to Reviewer mR8y**
>
> We sincerely appreciate the time you’ve dedicated to reviewing our paper, as well as your invaluable insights and support. Your positive feedback is highly motivating for us. Below, we address your primary concern and offer further clarification.
> > Q1. Is OpenCK the first combustion CFD benchmark? This is a strong statement. For example, Sandia's Engine Combustion Network by Pickett,Payri et al has been providing open data and benchamrking with regards to gasoline and diesel CFD since about 9 years ago. Another similar effort to this is BLASTNet in Chung et al (NeurIPS 2023) which involved Direct Numerical Simulation data of canonical combustion configurations. Please revise this statement to be more moderate.
>
> A1. Thank you for your comment. You have accurately identified an issue with our statement. The claim of "the first combustion CFD benchmark" should indeed be restricted to the field of fire. Accordingly, we have revised the sentence:
>
> "Open-CK is the first open-source benchmark dedicated to the study of combustion fluid dynamics, created through over 360 hours of numerical simulations supported by supercomputers."
>
> to:
>
> "Open-CK is the first open-source benchmark dedicated to the study of combustion fluid dynamics in the field of fire, created through over 360 hours of numerical simulations supported by supercomputers."
>
>
>
> > Q2. "Open-CK involves several PDEs" -- All listed PDE's are actually just scalar/vector conservation equations. Is this statement really true?
>
> A2. Thank you for your detailed comments. Please allow me to clarify that these four equations are not scalar/vector conservation equations.
>
> #### Navier-Stokes Equations:
> These are vector equations governing fluid flow, and they account for mass, momentum, and energy conservation in a more comprehensive way. Although they are vector equations, they are not simply conservation equations—they incorporate the full complexity of fluid dynamics, including viscosity, turbulence, and other complex phenomena.
>
> #### Energy Conservation Equation:
> This is not a simple scalar conservation equation but involves the distribution of energy across different forms, such as internal energy, kinetic energy, and thermal energy. The energy equation also incorporates terms for heat conduction, radiation, and convective heat transfer. These factors make it more complex than a straightforward scalar conservation equation. The basic form of the mass conservation equation is given by:
>
> $$
> \frac{\partial \rho}{\partial t} + \nabla \cdot (\rho \mathbf{v}) = 0
> $$
>
> Where:
> - $\rho$ is the fluid density,
> - $\mathbf{v}$ is the velocity vector,
> - $\nabla \cdot (\rho \mathbf{v})$ represents the divergence of the mass flux.
>
> This equation expresses the conservation of mass in a fluid, stating that the rate of change of mass within a control volume is equal to the net mass flux through the boundary of the control volume.
>
>
>
>
> #### Transport Equations for Smoke and Chemical Species:
> These involve both scalar and vector fields, as they describe the concentration of various species in the fluid flow. They account for the advection and diffusion of chemical species and smoke particles, and they are often coupled with reaction-diffusion terms that involve complex chemistry. This goes beyond a simple scalar conservation equation, as it involves multi-species transport and reactions.
>
> #### Heat Conduction Equation:
> While it may appear to be a scalar conservation equation, the heat conduction equation is often coupled with other equations (e.g., the Navier-Stokes and energy equations) and includes terms for heat sources, boundary conditions, and material properties, making it more complex than a simple scalar conservation law.
>
>
> > Q3. It would be interesting to see if the effects of model scaling has metrics in table 2. Bigger,expensive models tend to outperform smaller models. What does MSE vs FLOPS or MSE vs Params, SSIM vs (FLOPs, Params) look like in a scatter plot? This can provide more insight into useful architectures.
>
> A3. Thank you for your feedback. From the tabular data, it is unfortunate that the results do not support the claim that bigger, more expensive models tend to outperform smaller ones. Additionally, the scatter plots between pairs of metrics fail to reveal any meaningful insights. Therefore, we have decided not to include scatter plots in the original manuscript.

---

> > ### Author Response · Authors · 2024-11-25
> > **Kindly Request for Reviewer's Feedback**
> >
> > Dear Reviewer,
> >
> > Thank you so much for your time in improving our paper!
> >
> > Since the end of the rebuttal is coming soon, may we know if our response addresses your main concerns? If so, we kindly ask for your reconsideration of the score. Should you have any further advice, please let us know and we will be more than happy to engage in more discussion and improvements.

---

> > > ### Comment · Reviewer_mR8y · 2024-11-25
> > >
> > > Thank you for addressing Q1 and Q3 appropriately.
> > >
> > > I still think the PDEs mentioned are just mass, momentum and energy conservation, i.e. subsets of the same family of transport PDEs. So the claim that several PDEs are involved is a bit ambiguous.

---

> > > > ### Author Response · Authors · 2024-11-26
> > > > **Response to Reviewer mR8y**
> > > >
> > > > Dear Reviewer,
> > > >
> > > > Thank you for your valuable suggestions.
> > > >
> > > > After further research and discussion, we have decided to revise the relevant section of the paper to: 'Open-CK involves several partial differential equations (PDEs), including the Navier-Stokes equations (Li et al.; Takamoto et al., 2022), the Heat Conduction equation (Tieszen, 2001), and the Transport Equation for Smoke and Chemical Species (Drysdale, 2011).

---

> > > > > ### Comment · Reviewer_mR8y · 2024-11-26
> > > > >
> > > > > Thank you for addressing all my questions. Changing score to 8.

---

> ### Author Response · Authors · 2024-11-27
>
> Dear Reviewer mR8y
>
> We sincerely appreciate your valuable feedback and recognition. We are pleased to hear that your concerns have been addressed. We will certainly incorporate your suggestions into our revised version. Please do not hesitate to contact us if you have any further questions.
>
> Best regards,
>
> The Authors

---

### Official Review · Reviewer_V722 · 2024-11-04

**Soundness:** 2
**Presentation:** 2
**Contribution:** 3
**Rating:** 5
**Confidence:** 4

**Summary:**

This paper details a new dataset created using FDS for industrial fire parks. The authors use various state of the art SciML techniques on the dataset to show it's usefulness.

**Strengths:**

There is such a lack of data within the AI4Science domain, and in particular for fluid dynamics and fire modelling that any additional, well produced data is a strength. Clearly a lot of work has been undertaken to produce the data, which is a credit to the authors.

**Weaknesses:**

I'm afraid there are a number of weaknesses that makes this work, at present, not suitable for publication/presentation at ICLR:

1) Lack of any verification or validation of the underlying FDS methodology/mesh/technique for the simulated domain.
2) there are some phrases in the document sound odd i.e the use of the word supercomputers in italics. It sounds a little odd and is repeated in several places
3) no discussion on how the data will be maintained/stored/accessed
4) limited discussion of the limitations of the work

Overall, not high enough quality at present but I would encourage the authors to revisit and improve the paper for future conferences/publications.

**Questions:**

as per above

---

> ### Author Response · Authors · 2024-11-23
> **Response to Reviewer V722（part I）**
>
> We sincerely appreciate the time you’ve dedicated to reviewing our paper, as well as your invaluable insights and support. Your positive feedback is highly motivating for us. Below, we address your primary concern and offer further clarification.
>
> > Q1. Lack of any verification or validation of the underlying FDS methodology/mesh/technique for the simulated domain.
>
> A1. Thank you for your valuable feedback. We have conducted a grid independence analysis for a simulation domain of **150m × 150m × 10m** to validate the appropriateness of the grid size used in our numerical simulations. Larger grid sizes can compromise the accuracy of the simulation results, while smaller grid sizes may lead to excessive computational and time costs without significant benefits.
>
>
> The grid size is crucial in research outcomes that rely on numerical simulations. The ratio of fire characteristic diameter to grid size is the most widely used criterion in the literature to ensure the trustworthiness of the results. In the FDS user guide, $D^* / \delta x$ should range from 4 to 16 [1,2]. The characteristic diameter $D^*$ is given by:
>
> $$
> D^* = \left( \frac{Q}{\rho_\infty c_p T_\infty \sqrt{g}} \right)^{2/5}
> $$
>
> where:
> - $Q$ is the HRR (Heat Release Rate) of the fire source in kW
> - $\rho_\infty$ is air density in kg/m³
> - $c_p$ is air-specific heat in kJ/kg·K
> - $T_\infty$ is the ambient air temperature in K
> - $g$ is the gravitational constant in m/s²
>
> For example, when the HRR is set to **5 MW**, the calculation of $D^* = 1.826 \, \text{m}$ showed that the cell size varied from **0.114 m to 0.457 m**. Thus, for the mesh study, six grid sizes were used:
>
> - **0.500 m**
> - **0.250 m**
> - **0.167 m**
>
> In the experiment, we focus only on the physical fields within a low-altitude range of 10 meters. Therefore, the grid size for spaces beyond 10 meters is set arbitrarily, as it does not affect our simulation data. In this study, we set this value to 1 meter. All experiments demonstrate here are conducted on No.1 scenarios. The results are shown below.
>
> | grid sizes | number of cells | simulation time (s) |
> | - | - | - |
> | 0.500 m | 2749376 | 39520 |
> | 0.250 m | 16981440 | 358316 |
> | 0.167 m | 55463240 | 3219294 |
> However, it has been noted that when the mesh size is smaller than **0.5m**, there is little to no benefit and a significant increase in processing time. As a result, the grid resolution is chosen to be: **0.500 m**
>
>
>
> > Q2. there are some phrases in the document sound odd i.e the use of the word supercomputers in italics. It sounds a little odd and is repeated in several places
>
> A2. Thank you for your comment. In this study, we utilized a high-performance supercomputer due to the intensive computational demands of numerical simulations. Using standard desktop computers, such as personal PCs, would result in prohibitively long computation times. Supercomputers are the fastest high-performance systems available, and are distinguished from general-purpose computers by their processing power. Supercomputers can perform computations at hundreds of petaFLOPS, while desktop computers are limited to hundreds of gigaFLOPS to tens of teraFLOPS. The supercomputer, on the other hand, enables us to efficiently generate the initial dataset within a reasonable timeframe.
> > Q3. no discussion on how the data will be maintained/stored/accessed
>
> A3. Thanks for your feedback. For the ongoing maintenance of this dataset, we will conduct regular data quality checks to ensure its integrity and accuracy, removing duplicate, erroneous, or missing entries as needed. Additionally, we will use version control tools like Git to document the dataset's update history, enabling traceability and facilitating future research and reproducibility. An automated backup strategy will also be implemented, with data stored across both local servers and cloud platforms to safeguard against accidental loss.
> For the storage of this dataset, we utilize distributed storage to manage large-scale data while ensuring excellent scalability. Since the dataset consists entirely of numerical data, we adopt a standardized format (npy) to enhance compatibility and usability across different platforms.
> To facilitate efficient access and use of this dataset, we provide multiple query and download options through the links included in our documentation, enabling researchers to extract data based on their specific needs. Additionally, we offer comprehensive documentation and guides, including detailed descriptions of data fields, usage examples, and FAQs, ensuring users can quickly get started and accurately understand the dataset.

---

> ### Author Response · Authors · 2024-11-23
> **Response to Reviewer V722（part II）**
>
> > Q4. limited discussion of the limitations of the work
>
> A4. Thank you for your comment. We agree with your opinion, and therefore, in the "Future Insights & Limitations" section of the paper, we have expanded the discussion of the limitations of the work. The following points have been added:
> #### 1. Dependence on Simulated Data
>
> The Open-CK dataset is primarily generated through Computational Fluid Dynamics (CFD) simulations. While CFD simulations provide high-precision fire behavior predictions under controlled conditions, they cannot fully capture the complexities of real-world fires. Actual fires are influenced by numerous factors, such as environmental conditions, building materials, and airflow dynamics, which are often difficult to replicate entirely in simulations. This is particularly true in the early stages of a fire, where propagation and expansion are highly susceptible to unpredictable environmental changes. As a result, the simulated outcomes in the Open-CK dataset may fail to reflect sudden events or the nonlinear dynamics of fire development in real-world scenarios, potentially limiting the generalization capabilities of models trained on this dataset.
>
> #### 2. Limitations of Simulation Resolution and Accuracy
>
> Although Open-CK employs high-resolution CFD simulations, the accuracy of these simulations is constrained by the input parameters and available computational resources. CFD simulations demand substantial computational power and rely on precise physical models. Even the most advanced computational platforms face limitations in capturing every intricate detail. For instance, small-scale physical phenomena, such as localized airflow disturbances and subtle heat transfer processes, may not be adequately represented due to insufficient resolution or computational precision. Moreover, the accuracy of the input parameters is crucial; inaccuracies in these settings can lead to deviations in simulation results, ultimately impacting the predictive performance of models built on the dataset.
>
> #### 3. Limited Representation of Real-World Complexity
>
> Despite encompassing a wide range of fire scenarios, the Open-CK dataset cannot comprehensively replicate the complexity of real-world fires. Real fires are often influenced by unexpected factors, such as sudden changes in wind speed, structural variations in buildings, and unknown fire source locations. These factors significantly increase the unpredictability of fire progression. CFD simulations, however, are typically conducted under predefined conditions and environments, limiting their ability to account for these uncontrollable elements. In particular, the abrupt changes and long-term dynamics of real fires, such as rapid fire source expansion or dramatic environmental shifts, may not be accurately represented in simulations. This discrepancy can impact the practical applicability of models developed using the dataset.
>
> #### 4. Challenges in Multidisciplinary Data Integration
>
> The Open-CK dataset integrates multi-physics simulation data, providing a multidisciplinary platform for fire research. However, the coupling of physical models still poses challenges. Fires involve the interaction of various physical phenomena, such as airflow dynamics, heat transfer, and combustion processes, which often exhibit complex nonlinear feedback mechanisms. While CFD simulations attempt to combine these factors, the interplay among different physical phenomena is still not comprehensively or precisely represented in multi-physics datasets. Additionally, ensuring data consistency across different physical domains is a significant challenge. To achieve accurate modeling, it is essential to integrate data from diverse fields effectively and ensure temporal and spatial synchronization, which is critical for developing more precise fire prediction models.
>
> [1] K.B. McGrattan, R. McDermott, S. Hostikka, J.E. Floyd, Fire Dynamics Simulator (Version5) User’s Guide, National Institute of Standards and Technology, Gaithersburg, Maryland, 2010.
>
> [2] Chen, M., Li, H., Li, P., Ouyang, D., Weng, J., Wang, J., & Liu, H. (2021). Fireball modeling and thermal hazards analysis of leaked 1, 1-difluoroethane in fluorine chemical industry based on FDS. Journal of Thermal Analysis and Calorimetry, 146, 355-366.

---

> ### Author Response · Authors · 2024-11-26
> **Kindly Request for Reviewer's Feedback**
>
> Dear Reviewer,
>
> Thank you so much for your time in improving our paper!
>
> Since the end of the rebuttal is coming soon, may we know if our response addresses your main concerns? If so, we kindly ask for your reconsideration of the score. Should you have any further advice, please let us know and we will be more than happy to engage in more discussion and improvements.

---

> > ### Comment · Reviewer_V722 · 2024-11-26
> > **increased score**
> >
> > Thank you very much for responding with detailed feedback. I have increased my score from 3 to 5 to reflect this. I am still borderline on whether this paper is of suitable quality to be published but I will let the area chair and program chairs balance the different reviews on the final opinion.

---

> > > ### Author Response · Authors · 2024-11-26
> > >
> > > Dear Reviewer V722,
> > >
> > > We sincerely appreciate your valuable feedback and recognition!  We will definitely incorporate your suggestions into our revised version. Please kindly let us know if you have any questions further!
> > >
> > > Best regards,
> > >
> > > the Authors

---

### Meta-Review · Area_Chair_WjMh · 2024-12-22

**Metareview:**

The authors introduce a new large-scale dataset for modeling combustion kinetics (called Open-CK), and use this as a way to benchmark for scientific machine learning approaches. The dataset is created by running high-fidelity multi-physics fire dynamics simulations over digital CAD models of various industrial parks. The authors do a thorough job of describing the dataset, its design and construction methodology, its composition, as well as benchmarking most commonly used scientific ML approaches. I think this paper is well done and contributes a useful SciML dataset for a fairly unique domain (fire spread in complex built environments), and overall support acceptance.

**Additional Comments On Reviewer Discussion:**

Most reviews were on the borderline in the beginning and raised a few (mostly clarification) questions. The authors responded satisfactorily.

Some suggestions for improvement based on the back and forth: 300 samples seems a bit small to capture the full diversity of fire safety scenarios. This dataset is a good starting point but consider expanding to a richer set of configurations. Another suggestion: the paper can benefit from a fair bit of editing. Some reviewers already pointed out the odd use of the word `supercomputer'. Also: the second and third paragraphs of Section 3.1 are unnecessary in my opinion and can be removed. Please consider reflecting these (and the other reviewers' comments) while preparing the final version.

---

### Decision · Program_Chairs · 2025-01-22

Accept (Poster)